# SFBD-OMNI: BRIDGE MODELS FOR LOSSY MEASUREMENT RESTORATION WITH LIMITED CLEAN SAMPLES

**Haoye Lu[1,2], Yaoliang Yu[1,2] & Darren Lo[1]**
[1]School of Computer Science, University of Waterloo
[2]Vector Institute
{haoye.lu,yaoliang.yu,dlslo}@uwaterloo.ca

## ABSTRACT

In many real-world scenarios, obtaining fully observed samples is prohibitively expensive or even infeasible, while partial and noisy observations are comparatively easy to collect. In this work, we study distribution restoration with abundant noisy samples, assuming the corruption process is available as a black-box generator. We show that this task can be framed as a one-sided entropic optimal transport problem and solved via an EM-like algorithm. We further provide a test criterion to determine whether the true underlying distribution is recoverable under per-sample information loss, and show that in otherwise unrecoverable cases, a small number of clean samples can render the distribution largely recoverable. Building on these insights, we introduce SFBD-OMNI, a bridge model-based framework that maps corrupted sample distributions to the ground-truth distribution. Our method generalizes Stochastic Forward-Backward Deconvolution (SFBD; Lu et al., 2025) to handle arbitrary measurement models beyond Gaussian corruption. Experiments across benchmark datasets and diverse measurement settings demonstrate significant improvements in both qualitative and quantitative performance.

## 1 INTRODUCTION

Diffusion-based generative models (Sohl-Dickstein et al., 2015; Ho et al., 2020; Song et al., 2021a;b; 2023) have attracted growing interest and are now regarded as one of the most powerful frameworks for modelling high-dimensional distributions. They have enabled remarkable progress across various domains (Croitoru et al., 2023), including image (Ho et al., 2020; Song et al., 2021a;b; Rombach et al., 2022b), audio (Kong et al., 2021; Yang et al., 2023), and video generation (Ho et al., 2022). Today, most state-of-the-art image and video generative models are diffusion-based or their variants, such as flow matching (Lipman et al., 2023) and consistency models (Song et al., 2023).

While much of their success is attributed to stable training dynamics, diffusion models (DMs), like nearly all other generative frameworks, also depend on large collections of high-quality training data. In many practical domains, however, such data are costly or even infeasible to obtain, whereas large volumes of corrupted samples are readily available. For example, in medical imaging, acquiring cleaner X-ray scans requires higher radiation doses, which can endanger patient health (Seibert, 2008), making most available scans inherently noisy. Likewise, in ground-based astronomical imaging, clean deep-space observations demand long exposures under ideal atmospheric conditions, yet most telescope images are degraded by atmospheric turbulence, sensor noise, and light pollution (Chimitt & Chan, 2023).

Given this reality, a natural question arises: *With only a limited number of clean samples but an abundance of corrupted ones, can we train a model to recover the clean sample distribution?* Under suitable identifiability conditions on the corruption process, Bora et al. (2018) demonstrated that a generative model can indeed be trained using only corrupted samples, by leveraging the GAN framework (Goodfellow, 2016). Building on this idea and the remarkable success of diffusion models, subsequent works have sought to recover data distributions under specific corruption processes– for example, Ambient Diffusion for pixel masking (Daras & Dimakis, 2023), Tweedie Diffusion (Daras et al., 2023), and Stochastic Forward-Backward Deconvolution (SFBD, Lu et al. 2025) for

additive Gaussian noise. However, to the best of our knowledge, there is no existing framework that both accommodates general corruption processes and theoretical guarantees, while exploiting the advantages of diffusion models. A more detailed review of related literature is provided in Appx B.

In this work, we address this gap by proposing a principled framework for the distribution recovery problem through diffusion-based models. Instead of formulating distribution learning as a min-max game via the variational representation of the Kullback-Leibler (KL) divergence, as in GANs, we show that an alternative variational form, provided by the Donsker-Varadhan principle (Donsker & Varadhan, 1983), reveals the problem to be essentially equivalent to a one-sided entropic optimal transport objective. This reformulation naturally yields an alternative minimization pipeline that fully leverages the design advantages of diffusion-based models. Importantly, our approach avoids adversarial training, making it both simpler to implement and more stable in practice. Since the method can be viewed as a generalization of the SFBD algorithm, we refer to it as *SFBD-OMNI*.

Under suitable identifiability conditions on the corruption process, the proposed method is theoretically guaranteed to recover the ground-truth clean data distribution. For practical corruption processes that do not satisfy these conditions, we further show that the clean distribution can still be largely recovered when a limited number of clean samples are available, and we provide convergence guarantees for this setting. Since the proposed alternating minimization algorithm requires training a sequence of neural networks, we also introduce an online variant that enables end-to-end training, simplifying implementation and potentially accelerating convergence, while still preserving optimality guarantees. Empirical results corroborate our theoretical analysis, and experiments across benchmark datasets demonstrate significant and consistent improvements over strong baselines under diverse measurement settings. A key strength of SFBD-OMNI is its robustness in scenarios where the identifiability condition fails: by incorporating a small number of clean samples, the method is still able to effectively guide recovery toward the true data distribution. The implementation is available at https://github.com/watml/SFBD-omni.git.

## 2 PRELIMINARY

**Diffusion models and SFBD.** Diffusion models learn distributions by progressively corrupting data with Gaussian noise and then training a model to approximate the reverse process through successive denoising steps. Formally, given a distribution $\mu$ over $\mathbb{R}^d$, the forward process is governed by a stochastic differential equation (SDE):

$$\mathrm{d}\mathbf{x}_t = \mathrm{d}\mathbf{w}_t, \ \ \mathbf{x}_0 \sim \mu \tag{1}$$

where $\{\mathbf{w}_t\}_{t \in [0,T]}$ is the standard Brownian motion. Eq (1) induces a transition kernel $p_{t|s}(\mathbf{x}_t|\mathbf{x}_s) = \mathcal{N}(\mathbf{x}_0, (t-s)\mathbf{I})$ for $t \geq s \geq 0$. Let $p_t^\mu(\mathbf{x}_t) = \int p_{t|s}(\mathbf{x}_t|\mathbf{x}_0)\,\mu(\mathbf{x}_0)\,\mathrm{d}\mathbf{x}_0$ denote the marginal distribution of $\mathbf{x}_t$ (in particular, $p_0^\mu = \mu$). Anderson (1982) showed that the backward SDE can describe the time-reversed process corresponding to the forward SDE:

$$\mathrm{d}\mathbf{x}_t = -\mathbf{s}(\mathbf{x}_t, t)\mathrm{d}t + \mathrm{d}\bar{\mathbf{w}}_t, \ \mathbf{x}_\tau \sim p_\tau, \tag{2}$$

where $\tau > 0$, $\bar{\mathbf{w}}_t$ is standard Brownian motion in reverse time and $\mathbf{s}(\cdot, t) = \nabla \log p_t(\cdot)$ is the score function. In practice, the score can be efficiently approximated via a neural network $\mathbf{s}_{\boldsymbol{\theta}}$ trained by minimizing the conditional score matching loss $\mathcal{L}_{\mathrm{CSM}}(\mathbf{s}_{\boldsymbol{\theta}}, \mu)$ (Song et al., 2021b). Crucially, this reverse SDE induces transition kernels that coincide with the posterior of the forward process:

$$p_{s|t}^\mu(\mathbf{x}_s|\mathbf{x}_t) = \frac{p_{t|s}(\mathbf{x}_t|\mathbf{x}_s)\,p_s^\mu(\mathbf{x}_s)}{p_t^\mu(\mathbf{x}_t)}, \quad \text{for } s \leq t \text{ in } [0, \tau]. \tag{3}$$

Consequently, sampling from $p_{s|\tau}^\mu(\mathbf{x}_s \mid \mathbf{x}_\tau)$ can be carried out by integrating Eq (2) backward from $\mathbf{x}_\tau$ with $t = \tau$. In standard diffusion models, $\tau$ is chosen sufficiently large so that $p_\tau^\mu \approx \mathcal{N}(0, \tau\mathbf{I})$. Thus, sampling from the model amounts to drawing $\mathbf{x}_\tau \sim \mathcal{N}(0, \tau\mathbf{I})$ followed by $\mathbf{x}_0 \sim p_{0|\tau}^\mu(\mathbf{x}_0 \mid \mathbf{x}_\tau)$.

In contrast, SFBD (Lu et al., 2025) operates in the regime of finite $\tau$, specifically considering a Gaussian corruption process realized through the forward transition kernel $p_{\tau|0}(\mathbf{x}_\tau \mid \mathbf{x}_0)$. In particular, they assume access to a limited set of clean samples $\mathcal{E}_{\mathrm{clean}}$ and a large set of Gaussian corrupted ones $\mathcal{E}_{\mathrm{noisy}}$ obtained through this forward transition kernel. For a set of samples $\mathcal{E}$, let $p_\mathcal{E}$ denote the corresponding empirical distribution. Starting from a pretrained model $\mathbf{s}_{\boldsymbol{\theta}_0}$ by minimizing

$\mathcal{L}_{\text{CSM}}(\mathbf{s}_{\boldsymbol{\theta}}, \mathcal{E}_{\text{clean}})$, the algorithm proceeds as follows: for $k = 1, 2, \ldots, K$

$$\mathcal{E}_k \leftarrow \{\, \mathbf{x}_0 : \mathbf{y} \in \mathcal{E}_{\text{noisy}},\ \text{solve Eq (2) from } t = \tau \text{ to } 0,\ \text{with } \mathbf{x}_\tau = \mathbf{y} \text{ and } \mathbf{s} = \mathbf{s}_{\boldsymbol{\theta}_{k-1}}.\} \qquad (4)$$

$$\boldsymbol{\theta}_k \leftarrow \text{Continue training } \mathbf{s}_{\boldsymbol{\theta}_{k-1}} \text{ to obtain } \mathbf{s}_{\boldsymbol{\theta}_k} \text{ by minimizing } \mathcal{L}_{\text{CSM}}(\mathbf{s}_{\boldsymbol{\theta}}, \mathcal{E}_k) \qquad (5)$$

Lu et al. (2025) proved that as $K \to \infty$, $p_{\mathcal{E}_K}$ converges to the true distribution $p_{\text{data}}$ by analyzing the evolution of the underlying stochastic processes, leveraging the relation in Eq (3) for all $(s, t) \in [0, \tau]$. However, this relation is inherently tied to the Gaussian forward corruption process in Eq (1), which makes extending the approach to arbitrary corruption processes challenging.

Interestingly, the sampling step (4) essentially corresponds to drawing from $p_{0|\tau}^\mu$ with $\mu = p_{\mathcal{E}_{k-1}}$. This observation suggests that, rather than enforcing the posterior relation in (3) for all $(s, t) \in [0, \tau]$ and learning it via score function approximation in (2), it may be sufficient to train a model that learns only the posterior $p_{0|\tau}^\mu$. In this case, we may extend the forward kernel $p_{\tau|0}$ to arbitrary corruption processes. Indeed, when the corruption process satisfies suitable identifiability conditions, a generalized SFBD method can be employed to recover the data distribution, as we will show in Sec 5. We conclude this section by presenting a unified framework to learn $p_{0|\tau}^\mu$ with bridge models.

**Learning posterior distributions with bridge models.** Unlike standard diffusion models, which learn to transform Gaussian noise into data samples via the backward SDE (2), bridge models generalize this idea to transformations between arbitrary distributions (Lipman et al., 2023; Peluchetti, 2023; Zhou et al., 2024). Given paired samples $(\mathbf{x}, \mathbf{y}) \sim \pi(\mathbf{x}, \mathbf{y})$ from a joint distribution $\pi$, a bridge model constructs a distributional path connecting the $x$-marginal $\pi_x$ and the $y$-marginal $\pi_y$ by interpolating between each pair $(\mathbf{x}, \mathbf{y})$ through transition processes (Peluchetti, 2023). Typical choices include line segments in flow matching and rectified flow (Liu et al., 2022; Lipman et al., 2023), or Brownian bridges in DDBM and I2SB (Liu et al., 2023; Zhou et al., 2024). The resulting process defines a transition path distribution $p_{t|01}(\mathbf{x}_t \mid \mathbf{x}_0 = \mathbf{x}, \mathbf{x}_1 = \mathbf{y})$, whose evolution from $t = 1$ to 0 can often be expressed in closed form via a backward SDE (Peluchetti, 2023):

$$d\mathbf{x}_t = \mathbf{f}(\mathbf{x}_t; \mathbf{x}_0, \mathbf{x}_1, t)\, dt + g(t)\, d\bar{\mathbf{w}}_t. \qquad (6)$$

Let $\mathbf{f}_{\boldsymbol{\theta}}(\mathbf{x}_t; \mathbf{x}_1, t)$ be the minimizer of the conditional drift matching (CDM) loss

$$\mathcal{L}_{\text{CDM}}(\boldsymbol{\theta}, \pi) = \mathbb{E}_{t \sim \mathcal{U}}\, \mathbb{E}_{(\mathbf{x}_0, \mathbf{x}_1) \sim \pi}\, \mathbb{E}_{\mathbf{x}_t \sim p_{t|01}} \big\| \mathbf{f}(\mathbf{x}_t; \mathbf{x}_0, \mathbf{x}_1, t) - \mathbf{f}_{\boldsymbol{\theta}}(\mathbf{x}_t; \mathbf{x}_1, t) \big\|^2, \qquad (7)$$

where $\mathcal{U}$ is a sampling distribution over $t \in (0, 1)$. It then follows that samples from $\pi_{0|1}(\mathbf{x}_0 \mid \mathbf{y})$ can be obtained by integrating from $t = 1$ to 0 with $\mathbf{x}_1 = \mathbf{y}$ (Peluchetti, 2023; De Bortoli et al., 2023):[1]

$$d\mathbf{x}_t = \mathbf{f}_{\boldsymbol{\theta}}(\mathbf{x}_t; \mathbf{x}_1, t)\, dt + g(t)\, d\bar{\mathbf{w}}_t. \qquad (8)$$

In this way, given a Markov kernel $r(\mathbf{y} \mid \mathbf{x})$ for a general corruption process and a sample distribution $\mu$, the joint distribution of $(\mathbf{x}, \mathbf{y})$ is $\pi(\mathbf{x}, \mathbf{y}) = \mu(\mathbf{x})\, r(\mathbf{y} \mid \mathbf{x})$. A bridge model can then be trained to learn the posterior distribution in a manner analogous to diffusion models, using a CDM loss $\mathcal{L}_{\text{CDM}}$ corresponding to the chosen transition process.

## 3 KULLBACK–LEIBLER AMBIENT PROJECTION PROBLEM

Let $r(\cdot \mid \mathbf{x})$ denote the Markov kernel for the corruption process. Define the corresponding corruption operator $\mathcal{T}_r$, which maps a clean distribution $\mu$ to its corrupted counterpart:

$$\mathcal{T}_r \mu (\mathbf{y}) := \int r(\mathbf{y} \mid \mathbf{x})\, \mu(\mathbf{x})\, d\mathbf{x}. \qquad (9)$$

Given the corrupted data distribution $q := \mathcal{T}_r p_{\text{data}}$, our objective, following the classical GAN formulation in AmbientGAN (Bora et al., 2018), is to recover $p_{\text{data}}$ by solving

$$p^* = \arg\min_p D_{\text{KL}} \left( q \,\|\, \mathcal{T}_r p \right). \qquad (10)$$

The intuition is that minimizing the discrepancy between corrupted distributions drives $p$ toward the true clean distribution $p_{\text{data}}$. We refer to this optimization task as the Kullback–Leibler Ambient Projection (KLAP) problem.

---

[1]If $\mathbf{x}$ and $\mathbf{y}$ are connected by a deterministic path (i.e., $g = 0$ in Eq (6)), the sampling process may become ill-conditioned, as it degenerates to a deterministic mapping. To mitigate this, $\mathbf{y}$ can be perturbed with a small Gaussian noise during both training and sampling. See Appx C for details.

### 3.1 IDENTIFIABILITY

Whether the recovery is possible depends on the choice of the corruption kernel $r(\cdot \mid \mathbf{y})$. For instance, if $x$ is an image and $r(\cdot \mid \mathbf{y})$ always outputs a white patch, then the corrupted distribution $q = \mathcal{T}_r p_{\text{data}}$ collapses to a single point mass on the white patch, regardless of $p_{\text{data}}$. In this degenerate case, every distribution $p$ achieves the same objective value in (10), so the minimizer $p^*$ need not equal the true distribution $p_{\text{data}}$. The next proposition characterizes when minimizing (10) recovers $p^*$.

**Proposition 1** (Identifiability Condition). *Let $\mathcal{P}(X)$ denote the set of clean sample distributions. When the corruption kernel $r(\cdot \mid \mathbf{x})$ depends continuously on $\mathbf{x}$, the convex objective in Eq (10) admits a unique minimizer $p^* = p_{data}$ whenever $\mathcal{T}_r$ is injective on $\mathcal{P}(X)$. If $\mathcal{T}_r$ is not injective, the objective is still convex, but all distributions $p$ satisfying $\mathcal{T}_r p = \mathcal{T}_r p_{data}$ are minimizers.*

All proofs are deferred to the appendix. We highlight several common corruption operators $\mathcal{T}_r$ together with their injectivity properties:

*Additive noise.* If $\mathbf{y} = \mathbf{x} + \boldsymbol{\epsilon}$ with noise $\epsilon \sim \nu$, then $r(\mathbf{y} \mid \mathbf{x}) = \nu(\mathbf{y} - \mathbf{x})$. When $\nu$ has a characteristic function without zeros (e.g., Gaussian), the induced convolution operator $p \mapsto p * \nu$ is injective. This setting corresponds to the classical density deconvolution problem (Meister, 2009), with SFBD (Lu et al., 2025) addressing the Gaussian case in particular.

*Random dropout.* Each pixel is masked with probability $\alpha > 0$ and otherwise unchanged. It can be shown that when each pixel is masked independently, $\mathcal{T}_r$ is injective (Bora et al., 2018). (Non-injective when $\alpha = 1$.)

*Linear transforms.* If $\mathbf{y} = A\mathbf{x}$ for a linear map $A$, $r(\mathbf{y} \mid \mathbf{x}) = \delta(\mathbf{y} - A\mathbf{x})$. If $A$ has full column rank (hence is injective), then $\mathcal{T}_r$ is also injective. (Non-injective if $A$ has a nontrivial nullspace, such as projections or grayscale conversions of images.)

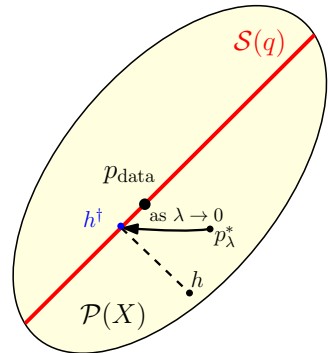

Figure 1: Effect of $\lambda$ on $p_\lambda^*$. As $\lambda \to 0$, the first term in Eq (11) ensures that $p$ remains within $\mathcal{S}(q)$, while the second term selects the element $h^\dagger \in S(q)$ closest to $h$. Consequently, $p_\lambda^*$ converges to $h^\dagger$, which represents the projection of $h$ onto the feasible set $\mathcal{S}(q)$.

### 3.2 AUGMENTED KLAP

As noted in Prop 1, if $\mathcal{T}_r$ is not injective, the objective is convex but not strictly convex. In this case, any distribution $p \in \mathcal{P}(X)$ with $\mathcal{T}_r p = \mathcal{T}_r p_{\text{data}}$ is a minimizer, and we denote this solution set by $\mathcal{S}(q)$. Thus, $p_{\text{data}}$ cannot be uniquely identified from the noisy distribution. One way to overcome this ambiguity is to incorporate additional information. In practice, this often comes from a small number of clean samples or, more generally, from a prior distribution $h$ over $p_{\text{data}}$. This motivates the following augmented formulation.

Given the corruption operator $\mathcal{T}_r$ defined in Eq (9), a *prior distribution* $h$ over $p_{\text{data}}$, and a *regularization parameter* $\lambda \geq 0$, we consider the following optimization problem:

$$p_\lambda^* = \arg\min_{p \in \mathcal{P}(X)} \mathcal{J}_\lambda(p), \quad \text{where} \quad \mathcal{J}_\lambda(p) := D_{\text{KL}}\left(q \parallel \mathcal{T}_r p\right) + \lambda D_{\text{KL}}\left(h \parallel p\right). \quad (11)$$

For $\lambda > 0$, the strict convexity of the second term ensures the entire objective is strictly convex with a unique minimizer $p_\lambda^*$, whereas for $\lambda = 0$ it reduces to the classical ambient problem.

For intuition, consider a corruption process $r$ that maps colour images to grayscale, with $p_{\text{data}}$ consisting of human face images. Here $\mathcal{T}_r$ is not injective, since many different colourings yield the same grayscale distribution. In other words, $\mathcal{S}(q)$ contains multiple elements. Thus, when $\lambda = 0$, we can recover the distribution of face structures but not the true colour patterns. To capture the full colour distribution, we may assume access to a few clean colour images from $p_{\text{data}}$ and encourage $p$ to align with their empirical distribution $h$ by choosing $\lambda > 0$.

Fig 1 illustrates how the additional regularization term shapes the optimal solution. As $\lambda \to 0$, the first term in Eq (11) keeps $p$ within $\mathcal{S}(q)$, while the second selects the element $h^\dagger \in \mathcal{S}(q)$ closest to $h$. We formalize this observation in the following proposition.

**Proposition 2.** *Let $h^{\dagger} = \arg\min_{p \in \mathcal{S}(q)} D_{\mathrm{KL}}(h \parallel p)$ denote the Information-projection of $h$ onto the original KLAP solution set. Then the minimizer of Eq (11), $p_{\lambda}^{*}$, converges to $h^{\dagger}$ as $\lambda \to 0$.*

**Clean samples also matter under injective $\mathcal{T}_r$ – Identifiability $\neq$ Recoverability.** While Prop 1 shows that if $\mathcal{T}_r$ is injective, then $p_{\mathrm{data}}$ is in principle recoverable by minimizing Eq (10), this guarantee relies on having access to the true corrupted density $q = \mathcal{T}_r p_{\mathrm{data}}$. In practice, however, $q$ must be estimated from finitely many noisy samples, and the resulting estimation error is amplified through the inverse of $\mathcal{T}_r$. Consequently, the minimizer of Eq (10) based on an empirical estimate of $q$ can deviate substantially from $p_{\mathrm{data}}$. For additive-noise corruption operators $\mathcal{T}_r$, the unfavourable sample complexity of this inverse problem is well documented in the density deconvolution literature (see, e.g., Meister (2009)), and the pessimistic rates suggest that acquiring enough noisy samples to train a high-quality model is often practically infeasible (Lu et al., 2025). To overcome this issue, in Sec 6, we show that even a very small number of clean samples (as few as 50) can substantially mitigate this difficulty, consistent with the findings of Lu et al. (2025).

## 4 Two Variational Perspectives of KLAP

In this section, we present two variational perspectives for characterizing KLAP, each derived from a different variational formulation of the KL divergence. The first perspective corresponds to the classical formulation, which was previously employed in training Ambient GANs (Bora et al., 2018), and is included here for completeness. The second perspective reveals that the classical KLAP can be viewed as a one-sided entropic optimal transport (OT) problem and also leads to an alternative minimization algorithm for solving both the classical and augmented KLAP formulation.

### 4.1 Ambient GAN's Formulation

For any convex function $f$, a corresponding $f$-divergence can be defined: $D_f(q\|m) = \int m(\mathbf{y}) f(\frac{q(\mathbf{y})}{m(\mathbf{y})}) \mathrm{d}\mathbf{y}$ (Nowozin et al., 2016), which also admits an variational form

$$D_f(q\|m) = \max_g \left\{ \mathbb{E}_q[g(Y)] - \mathbb{E}_m[f^*(g(Y))] \right\}, \tag{12}$$

where $f^*$ is the convex conjugate of $f$. When $f(x) = x \ln x$, $D_f$ reduces to the KL divergence. As a result, with this choice of $f$, the original KLAP problem (10) can be rewritten as

$$\min_p D_f(q \parallel \mathcal{T}_r p) = \min_p \max_g \left\{ \mathbb{E}_q[g(Y)] - \mathbb{E}_{\mathcal{T}_r p}\left[f^*(g(Y))\right] \right\}. \tag{13}$$

This min-max formulation can be naturally implemented in the standard GAN framework (Goodfellow, 2016), with $g$ as the discriminator and $p$ parameterized by the generator. Bora et al. (2018) showed that this setup can recover $p_{\mathrm{data}}$ when $\mathcal{T}_r$ is injective and the corruption process is differentiable with respect to the clean inputs.

To the best of our knowledge, existing KLAP-based frameworks cannot directly incorporate the additional identifiability term or support a more scalable, diffusion/bridge-style generator. In Sec 4.2, we introduce an alternative variational formulation that yields an alternating-minimization algorithm (Sec 5) addressing both issues. Notably, the method requires only black-box access to the corruption process, without any differentiability assumptions.

### 4.2 One-sided Entropic Optimal Transport Formulation

Let $f_{\mathbf{y}}(\mathbf{x}) = \log r(\mathbf{y} \mid \mathbf{x})$.[2] Rather than invoking the variational representation of KL-divergence, we apply the Donsker-Varadhan variational principle (Donsker & Varadhan, 1983):

$$\log \mathbb{E}_{\mathbf{x} \sim p}\left[e^{f_{\mathbf{y}}(\mathbf{x})}\right] = \max_{u_{\mathbf{y}}} \mathbb{E}_{\mathbf{x} \sim u_{\mathbf{y}}}[f_{\mathbf{y}}(\mathbf{x})] - D_{\mathrm{KL}}(u_{\mathbf{y}} \parallel p), \tag{14}$$

where $u_{\mathbf{y}}$ denotes a distribution of $\mathbf{x}$ given $\mathbf{y}$. Taking expectation over $\mathbf{y} \sim q$ and rearranging yields

$$D_{\mathrm{KL}}(q \parallel \mathcal{T}_r p) = \min_{u_{\mathbf{y}}} \mathbb{E}_{\mathbf{y} \sim q}\left[D_{\mathrm{KL}}(u_{\mathbf{y}} \parallel p) - \mathbb{E}_{\mathbf{x} \sim u_{\mathbf{y}}}[f_{\mathbf{y}}(\mathbf{x})]\right] + C, \tag{15}$$

where $C$ collects the terms independent of $p$ (see Appx E for the derivation). As a result, the augmented KLAP problem (11) is equivalent to

$$\arg\min_p \min_{u_{\mathbf{y}}} \mathcal{F}_{\lambda}(p, u_{\mathbf{y}}) \tag{16}$$

---

[2]We assume $r(\cdot \mid \mathbf{x})$ has full support; this can be enforced by injecting an infinitesimal Gaussian noise to $\mathbf{y}$.

with

$$\mathcal{F}_\lambda(p, u_\mathbf{y}) \coloneqq \mathbb{E}_{y \sim q}\big[D_{\mathrm{KL}}\left(u_\mathbf{y} \parallel p\right) - \mathbb{E}_{\mathbf{x} \sim u_\mathbf{y}}[f_\mathbf{y}(\mathbf{x})]\big] + \lambda D_{\mathrm{KL}}\left(h \parallel p\right).$$

This nested minimization suggests an alternative strategy for solving the (augmented) KLAP problem in Sec 5. We conclude this section by noting that this observation allows KLAP to be viewed as a variant of classical entropic OT, offering a new perspective to understand the KLAP problem.

**Proposition 3.** *Define the cost function $c(\mathbf{x}, \mathbf{y}) \coloneqq -\log r(\mathbf{y} \mid \mathbf{x})$. Problem (16) is equivalent to*

$$\arg\min_p \Phi(p) + \lambda\, D_{\mathrm{KL}}\left(h \parallel p\right)$$

*with*

$$\Phi(p) \coloneqq \min_{\pi \in \Pi_\mathbf{y}(q)} \iint c(\mathbf{x}, \mathbf{y})\,\pi(\mathbf{x}, \mathbf{y})\,\mathrm{d}\mathbf{x}\mathrm{d}\mathbf{y} + D_{\mathrm{KL}}\left(\pi \parallel p \otimes q\right)$$

*where $\Pi_\mathbf{y}(q)$ denotes the set of joint distributions of $(\mathbf{x}, \mathbf{y})$ with $\mathbf{y}$-marginal fixed to $q$. Moreover, when $\lambda = 0$, the optimal solution $p^*$ coincides with the $\mathbf{x}$-marginal of the corresponding minimizer $\pi^*$ in the inner problem.*

Notably, $\Phi(p)$ in Prop 3 coincides with the entropic OT objective (Cuturi, 2013), but with constraints imposed only on the $\mathbf{y}$-marginal rather than on both marginals. In particular, when the cost function is quadratic, as in the case of a Gaussian corruption kernel, the optimal coupling $\pi^*$ corresponds to the Schrödinger Bridge (Léonard, 2014). Moreover, Prop 3 shows that in the absence of the regularization toward the prior distribution $h$ (i.e., when $\lambda = 0$), the optimal solution $p^*$ induces an optimal coupling $\pi^*$ in the inner entropic OT problem whose marginals are precisely $p^*$ and $q$. This interpretation suggests that solving KLAP amounts to finding a distribution $p$ that minimizes the transportation cost induced by the corruption kernel, subject to entropy regularization. The methods introduced in Sec 5 provide an effective approach for solving this one-sided entropic OT problem.

## 5 STOCHASTIC FORWARD-BACKWARD DECONVOLUTION-OMNI

The variational formulation of the augmented KLAP in Eq (16) suggests an alternative minimization approach for finding the minimizer $p_\lambda^*$ defined in Eq (11). This leads to an algorithm that generalizes SFBD (Lu et al., 2025) to arbitrary corruption models, which we call *SFBD-OMNI*.

**SFBD-OMNI.** Starting from an arbitrary initialization $p^0(\mathbf{x})$, we minimize $\mathcal{F}(p, u_\mathbf{y})$ in (16) by alternating updates over $p$ and $u_\mathbf{y}$, holding the other fixed. Specifically, at each iteration, we compute

$$u_\mathbf{y}^k = \arg\min_{u_\mathbf{y}}\ \mathcal{F}_\lambda(p^k, u_\mathbf{y}), \qquad p^{k+1} = \arg\min_p\ \mathcal{F}_\lambda(p, u_\mathbf{y}^k), \qquad (17)$$

where both subproblems admit closed-form solutions:

$$u_\mathbf{y}^k(x) = \frac{p^k(x)\,r(\mathbf{y} \mid \mathbf{x})}{\mathcal{T}_r p^k(\mathbf{y})}, \qquad p^{k+1}(x) = \frac{1}{1+\lambda}\,\tilde{p}^{k+1}(x) + \frac{\lambda}{1+\lambda}\,h(x), \qquad (18)$$

with $\tilde{p}^{k+1}(\mathbf{x}) = \int q(\mathbf{y})\,u_\mathbf{y}^k(\mathbf{x})\,\mathrm{d}\mathbf{y}$.

Note that $u_\mathbf{y}^k$ is the posterior distribution of $p^k(x)$ under the joint distribution $\pi(\mathbf{x}, \mathbf{y}) = p^k(\mathbf{x})\,r(\mathbf{y}|\mathbf{x})$. As described in Sec 2, by introducing a transition process connecting $\mathbf{x}$ and $\mathbf{y}$, we can leverage a bridge model to learn this posterior in a manner analogous to diffusion models, by minimizing the corresponding CDM loss $\mathcal{L}_{\mathrm{CDM}}$. Let $u_{\boldsymbol{\theta}}$ denote the learnt posterior distribution. The quantity $\tilde{p}_k$ is then approximated using samples from $u_{\boldsymbol{\theta}}(\cdot \mid \mathbf{y})$ with $\mathbf{y} \sim q(\mathbf{y})$.

We describe the implementation of SFBD-OMNI in Alg 1, assuming access to a small set of clean samples that define the prior $h$, denoted $h_{\mathrm{clean}}$, which also serves as the initialization $p^0$. During training, $\tilde{p}^k$ is approximated by $p_\mathcal{E}$ and updated iteratively, while the mixture of $p_\mathcal{E}$ and $h_{\mathcal{E}_{\mathrm{clean}}}$ is realized through a weighted sampler.

**Online SFBD-OMNI.** The implementation of Alg 1 alternates between training and sampling, which in practice demands considerable manual intervention. Moreover, because $\mathcal{E}$ changes drastically at each iteration, optimizers such as Adam (Kingma & Ba, 2015) must be reset after every fine-tuning step; otherwise, stale momentum can trigger a sharp and irreversible increase in training loss. To

---

**Algorithm 1** SFBD-OMNI

**Input:** clean data $\mathcal{E}_{\text{clean}} = \{\mathbf{x}^{(i)}\}_{i=1}^{M}$, noisy data $\mathcal{E}_{\text{noisy}} = \{\mathbf{y}^{(i)}\}_{i=1}^{N}$, CDM loss $\mathcal{L}_{\text{CDM}}$

```
// Pretrain using clean samples
```
1 $\boldsymbol{\theta} \leftarrow$ Minimizing $\mathcal{L}_{\text{CDM}}\big(\boldsymbol{\theta}, h_{\mathcal{E}_{\text{clean}}}(\mathbf{x})\, r(\mathbf{y}|\mathbf{x})\big)$

2 $\mathcal{E} \leftarrow \{\mathbf{x}^{(i)} :$ take one sample from $u_{\boldsymbol{\theta}}(\mathbf{x}|\mathbf{y})$ for each corrupted sample $y \in \mathcal{E}_{\text{noisy}}\}$.

```
// Iteratively optimize with
   corrupted samples
```
3 **for** $k = 1, 2, \ldots, K$ **do**

4    $\boldsymbol{\theta} \leftarrow$ Minimizing $\mathcal{L}_{\text{CDM}}\big(\boldsymbol{\theta}, p(\mathbf{x})\, r(\mathbf{y}|\mathbf{x})\big)$ with $p = \frac{1}{1+\lambda} p_{\mathcal{E}} + \frac{\lambda}{1+\lambda} h_{\mathcal{E}_{\text{clean}}}$.

5    $\mathcal{E} \leftarrow \{\mathbf{x}^{(i)} :$ take one sample from $u_{\boldsymbol{\theta}}(\mathbf{x}|\mathbf{y})$ for each corrupted sample $\mathbf{y} \in \mathcal{E}_{\text{noisy}}\}$

**Output:** Final $u_{\boldsymbol{\theta}}$

---

**Algorithm 2** Online SFBD-OMNI

**Input:** clean data $\mathcal{E}_{\text{clean}} = \{\mathbf{x}^{(i)}\}_{i=1}^{M}$, noisy data $\mathcal{E}_{\text{noisy}} = \{\mathbf{y}^{(i)}\}_{i=1}^{N}$, gradient steps $m$, CDM loss $\mathcal{L}_{\text{CDM}}$

```
// Pretrain using clean samples
```
1 $\boldsymbol{\theta} \leftarrow$ Minimizing $\mathcal{L}_{\text{CDM}}\big(\boldsymbol{\theta}, h_{\mathcal{E}_{\text{clean}}}(\mathbf{x})\, r(\mathbf{y}|\mathbf{x})\big)$

2 $\mathcal{E} \leftarrow \{\mathbf{x}^{(i)} :$ take one sample from $u_{\boldsymbol{\theta}}(\mathbf{x}|\mathbf{y})$ for each corrupted sample $\mathbf{y} \in \mathcal{E}_{\text{noisy}}\}$.

```
// Iteratively optimize with
   corrupted samples (online
   updates)
```
3 **for** $k = 1, 2, \ldots, K$ **do**

4    $\boldsymbol{\theta} \leftarrow$ Minimizing $\mathcal{L}_{\text{CDM}}\big(\boldsymbol{\theta}, p(\mathbf{x})\, r(\mathbf{y}|\mathbf{x})\big)$ with $p = \frac{1}{1+\lambda} p_{\mathcal{E}} + \frac{\lambda}{1+\lambda} h_{\mathcal{E}_{\text{clean}}}$.

5    $\mathcal{E} \leftarrow \{$Replace ratio $\gamma$ of samples in $\mathcal{E}$ with the new ones by sampling $\mathbf{x}$ from $u_{\boldsymbol{\theta}}(\mathbf{x}|\mathbf{y})$ for $\mathbf{y}$ drawn from $\mathcal{E}_{\text{noisy}}\}$

**Output:** Final $u_{\boldsymbol{\theta}}$

---

guarantee convergence in each iteration, the network must also be optimized for a sufficiently large number of steps. However, this can lead to overfitting on the current iterate $p^k$, making subsequent adaptation to new targets more difficult.

To address these challenges, we introduce an online variant in Alg 2, where a fraction $\gamma$ of the reconstructed set $\mathcal{E}$ is refreshed at each iteration. This corresponds to updating $\tilde{p}^{k+1}(\mathbf{x})$ in Eq (18) as

$$\tilde{p}^{k+1}(\mathbf{x}) = \gamma \int q(\mathbf{y})\, u_{\mathbf{y}}^k(\mathbf{x})\, \mathrm{d}\mathbf{y} + (1-\gamma)\, \tilde{p}^k(\mathbf{x}) \quad \text{with} \quad \tilde{p}^0(\mathbf{x}) = \int q(\mathbf{y})\, u_{\mathbf{y}}^0(\mathbf{x})\, \mathrm{d}\mathbf{y}. \quad (19)$$

When $\gamma = 1$, the algorithm reduces to the standard SFBD-OMNI. Because $\mathcal{E}$ changes only slightly after each update, we can continue optimizing $u_{\boldsymbol{\theta}}$ for additional gradient steps without resetting the optimizer state, allowing it to adapt smoothly to the new minimum. This strategy reduces manual intervention and accelerates convergence. In Prop 4, we show that this "lazy" update scheme still guarantees convergence to the optimum. Since the result covers the case $\gamma = 1$, it also establishes the convergence of SFBD-OMNI.

**Proposition 4** (Convergence to the optimum). *Let the distribution sequences $\{u_y^k\}$ and $\{p^k\}$ evolve according to Eq (18), with $\tilde{p}^k$ updated by Eq (19). Starting from an arbitrary initialization $p^0$ and for $\gamma \in (0, 1]$, under mild assumptions, we have $p^k \to p_\lambda^*$ as $k \to \infty$. Moreover, when $\lambda \to 0$, we have*

$$\lim_{k \to \infty} p^k = h^\dagger, \quad D_{\text{KL}}\big(h^\dagger \,\|\, p^{k+1}\big) \le D_{\text{KL}}\big(h^\dagger \,\|\, p^k\big). \quad (20)$$

*In addition, the following bounds hold:*

$$\min_{1 \le k \le K} D_{\text{KL}}\big(q \,\|\, \mathcal{T}_r p^k\big) \le \frac{D_{\text{KL}}\big(h^\dagger \,\|\, p^0\big)}{\gamma K}, \quad (21)$$

*where $K$ denotes the total number of iterations and $q = \mathcal{T}_r p_{data}$.*

While Eq (21) may suggest that a smaller $\gamma$ leads to slower convergence, note that with smaller $\gamma$, the set $\mathcal{E}$ is only partially updated, so the network $u_{\boldsymbol{\theta}}$ requires fewer steps to converge. Thus, although a larger $K$ may be needed to guarantee convergence, each step is cheaper, and the total training time does not necessarily increase. In practice, since the optimizer does not need to reset, training time can even decrease.

**Comparison to existing methods.** When $\lambda = 0$ and the corruption process is Gaussian noise injection, with the posterior modeled via the backward SDE in Sec 2, our framework reduces to SFBD (Lu et al., 2025). In EMDiffusion, Bai et al. (2024) heuristically derive an iterative rule that coincides with SFBD-OMNI's update in Eq (18) when $\lambda = 0$; our work establishes convergence of

| Method | CIFAR-10 | | | CelebA | |
| --- | --- | --- | --- | --- | --- |
| | Pixel Masking (✓) | Additive Gauss. (✓) | Grayscale (✗) | Gauss. Blur (✗) | Grayscale (✗) |
| Noise2Self (Batson & Royer, 2019) | – | 92.06 | – | – | – |
| SURE-Score (Aali et al., 2023) | 220.01 | 132.61 | 109.04 | 191.96 | 219.81 |
| AmbientDiff (Daras & Dimakis, 2023) | 28.88 | – | – | – | – |
| EMDiffusion (Bai et al., 2024) | **21.08** | 86.47 | 115.11 | 91.89 | 59.04 |
| SFBD (Lu et al., 2025) | – | 13.53 | – | – | – |
| SFBD-OMNI (**ours**) | 21.31 | **10.81** | 32.61 | 11.60 | 11.85 |
| Online SFBD-OMNI (**ours**) | 22.43 | 11.06 | **31.32** | **10.28** | **11.21** |

Table 1: FID scores across different corruption processes on CIFAR-10 and CelebA. Processes marked with ✓ satisfy the identifiability condition, while those marked with ✗ do not. Pixel masking is applied with probability $p = 0.6$ per pixel. Additive Gaussian corruption adds noise with $\sigma = 0.2$ to each clean sample. The grayscale process converts a color image into a single-channel grayscale image, while Gaussian blur is applied with a kernel size of nine and $\sigma = 2$. All methods, except Noise2Self, are pretrained on 50 clean images randomly sampled from the training dataset.

this rule to the optimal solution, which EMDiffusion does not, and further extends it with an online formulation and the ability to handle non-identifiable corruption processes. Unlike AmbientGAN (Bora et al., 2018), which requires differentiating noisy samples with respect to clean ones and cannot address non-identifiable corruption processes, SFBD-OMNI and the online version assume black-box access to the corruption process, avoid adversarial training, and thus sidestep common issues such as gradient vanishing (Goodfellow et al., 2014; Miyato et al., 2018; Fedus et al., 2018) and mode collapse (Goodfellow, 2016; Arjovsky & Bottou, 2017; Mescheder et al., 2018).

## 6 EMPIRICAL STUDY

In this section, we evaluate the proposed SFBD-OMNI framework introduced in Sec 5. Across diverse benchmark settings, both SFBD-OMNI and its online variant demonstrate superior performance over existing approaches for recovering the original data distribution from corrupted observations. Furthermore, our ablation studies show that the method can effectively address non-identifiable corruption processes.

**Datasets and evaluation metrics.** Our experiments are performed on CIFAR-10 (Krizhevsky & Hinton, 2009) and CelebA (Liu et al., 2022), with image sizes of $32 \times 32$ and $64 \times 64$, respectively. CIFAR-10 contains 50,000 training samples and 10,000 test samples spanning 10 object categories. CelebA is a large-scale dataset of human faces with a standard split of 162,770 training, 19,867 validation, and 19,962 test images. For CelebA, preprocessing follows the official tool released with DDIM (Song et al., 2021a).

**Models and other configurations.** In our implementation, we parameterize $u_{\boldsymbol{\theta}}(\mathbf{x} \mid \mathbf{y})$ with a flow-matching model (Lipman et al., 2023) and apply small endpoint perturbations to $\mathbf{y}$ to avoid degeneracy, as described in Appx C. We adopt flow matching because it converges faster and has a lower-variance training objective than diffusion-based models, while achieving comparable or even superior sample quality. This computational efficiency is particularly important in our framework, where the bridge models are trained repeatedly against a moving target distribution. To further mitigate overfitting, we adopt the non-leaky augmentation technique (Karras et al., 2022). For the classical SFBD-OMNI, after pretraining on a small set of clean samples, we set the clean-sample weight $\frac{\lambda}{1+\lambda}$ to zero when the corruption process satisfies the identifiability condition; otherwise, we use $\frac{\lambda}{1+\lambda} = 0.2$, unless specified otherwise. For the flow variant, we fix $\frac{\lambda}{1+\lambda} = 0.2$, as this setting yields more stable training. In addition, unless noted, we set the noisy-set update ratio to $\gamma = 0.002$ and perform the update at the end of each training epoch. For sampling, we generate samples by first picking $\mathbf{y}$ from the noisy dataset and then sampling from the final $\mathbf{u}_{\boldsymbol{\theta}}(\mathbf{x} \mid \mathbf{y})$. Additional training configurations are provided in Appx G. We evaluate image quality using the Frechet Inception Distance (FID), computed between the reference dataset and 50,000 images generated by the models.

**Performance comparison.** In Table 1, we compare SFBD-OMNI with representative models trained on noisy images corrupted by various processes. As discussed in Sec 3.1, pixel masking and additive Gaussian noise satisfy the identifiability condition, making it theoretically possible to recover the data distribution using only noisy samples. In contrast, grayscale conversion and Gaussian blur do not satisfy this condition, meaning that additional prior information is required for effective distribution recovery. (Notably, Gaussian blur discards high-frequency components of an image and can be viewed as a projection in the Fourier domain.)

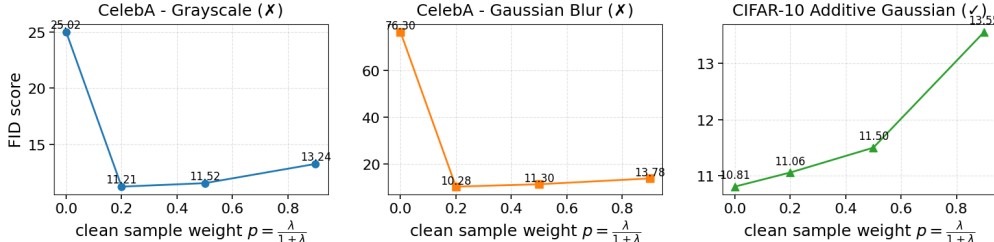

Figure 2: FID scores of Online SFBD-OMNI under different clean sample weights $p = \frac{\lambda}{1+\lambda}$ across various corruption processes. Processes marked with ✓ satisfy the identifiability condition, while those marked with ✗ do not.

For the baseline models, Noise2Self (Batson & Royer, 2019) is a general-purpose denoising method trained with self-supervised techniques. SURE-Score (Aali et al., 2023) and EMDiffusion (Bai et al., 2024) address general inverse problems, leveraging Stein's unbiased risk estimate and expectation–maximization, respectively. Notably, the update rules of EMDiffusion coincide with those of standard SFBD-OMNI when no additional prior information is incorporated, rendering it ineffective for non-identifiable corruption processes. AmbientDiff (Daras & Dimakis, 2023), in contrast, is specifically designed to train diffusion models on images corrupted by masking. We also report results from the original SFBD, which is tailored to additive Gaussian noise (Lu et al., 2025). (A discussion and empirical comparison with a very recent work, Ambient Diffusion OMNI (Daras et al., 2025b), is provided in Appx I.) Following Bai et al. (2024), unless otherwise stated, all methods except Noise2Self are pretrained on 50 clean images randomly sampled from the training dataset. In SFBD-OMNI and the flow variant, these images are further used as prior information during sequential training whenever the clean-sample weight $\frac{\lambda}{1+\lambda} > 0$. For all reported results, we consistently use the same set of 50 clean images.

As shown in Table 1, apart from the pixel masking corruption process, SFBD-OMNI and its flow variant consistently outperform the baselines, achieving substantially better performance on the non-identifiable processes. In the pixel masking case, EMDiffusion reports a marginally lower FID than SFBD-OMNI; however, the difference is negligible, indicating that SFBD-OMNI performs on par with EMDiffusion in this setting. For the non-identifiable corruptions, we observe that incorporating prior information, by jointly training the model with reconstructed samples in $\mathcal{E}$ and clean samples, effectively guides the model toward the true data distribution, as reflected in the much lower FID scores. In addition, because the flow-variant implementation always assigns a non-zero weight $\frac{\lambda}{1+\lambda}$ to clean samples for added stability, its optimal solution $p_\lambda^*$ deviates from the true data distribution in identifiable cases, leading to a slightly higher FID than classical SFBD-OMNI. In contrast, for the non-identifiable processes, this additional regularization is essential and applied in both variants. Consequently, the smooth updates and end-to-end training pipeline of the flow model provide it with an additional advantage, enabling it to achieve lower FID scores.

**Effect of the clean sample weights.** To examine how SFBD-OMNI leverages clean samples to mitigate identifiability issues, Fig 2 reports FID curves under varying clean-sample weights and corruption types (settings follow Table 1). When identifiability does not hold, using clean samples as a soft prior constraint guides the model toward the correct distribution; however, overly large weights pull the solution away from the target, increasing FID. Conversely, when identifiability is satisfied, this regularization is unnecessary and may even degrade performance. This phenomenon corroborates our discussion in Sec 3 and Sec 4. In particular, in identifiable setups, clean samples mainly help initialize $p_0$, after which training proceeds best without them (e.g., CIFAR-10 with Gaussian noise). When identifiability fails, clean samples must remain active ($\lambda > 0$) to avoid convergence to an arbitrary element of $\mathcal{S}(q)$, as seen in CelebA with Grayscale and Gaussian Blur, where removing clean samples increases FID dramatically. These trends align directly with the theoretical role of identifiability. Since the clean samples are only used for initializing $p_0$ when the identifiability condition is satisfied, we show in Appx J that it is acceptable to use samples from a similar distribution instead when clean samples are not available.

**Effect of the number of clean samples.** Fig 3a reports the FID scores of Online SFBD-OMNI on CelebA under Grayscale corruption for different amounts of clean data. Increasing the number of clean samples improves performance at both the pretraining and iterative optimization stages, though with diminishing returns. This is expected and aligned with our discussions in Sec 5: more clean

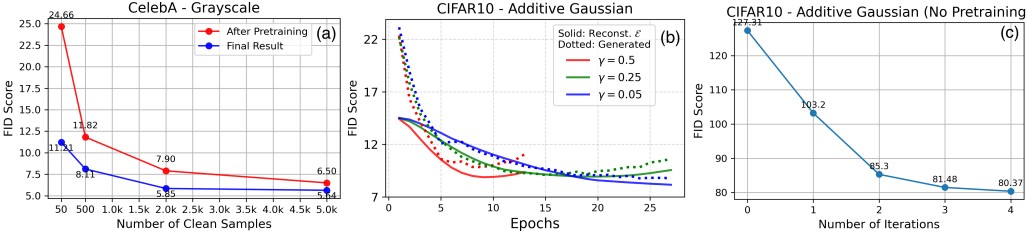

Figure 3: FID scores of SFBD-OMNI under different settings. (a) Online SFBD-OMNI FIDs under grayscale corruption for varying numbers of clean samples. (b) FID trajectories of the online version under additive Gaussian corruption ($\sigma = 0.5$) with 2k clean samples, for both the running reconstructed set $\mathcal{E}$ and a newly generated sample set. (c) FIDs of the classical SFBD-OMNI under additive Gaussian ($\sigma = 0.2$) without clean samples; iteration 0 represents the untrained model.

samples make the empirical clean distribution $h_{\text{clean}}$ closer to $p_{\text{data}}$, yielding a better initialization $p^0 = h_{\text{clean}}$ and a limiting distribution $p_\lambda^\star$ (defined in Eq (11)) that more closely matches $p_{\text{data}}$. Once $h_{\text{clean}}$ is already a good approximation, however, additional samples provide only marginal benefit.

**Effect of the update ratio $\gamma$.** Fig 3b shows the FID trajectories of both the running reconstructed sample set $\mathcal{E}$ and a newly generated sample set during the iterative optimization stage of Online SFBD-OMNI, evaluated under different reconstructed-sample update ratios $\gamma$. The experiment is conducted on CIFAR-10 with additive Gaussian corruption ($\sigma = 0.5$) and 2,000 clean samples. A larger $\gamma$ causes the reconstructed set $\mathcal{E}$ to be refreshed more frequently, which yields a sharper early decrease in FID (as seen for $\gamma = 0.5$). Yet, because $\mathcal{E}$ changes so rapidly, the model cannot fully adjust to the current reconstruction set before it is updated again. This instability appears as a growing discrepancy between the FIDs of reconstructed and newly generated samples after epoch 6, eventually degrading reconstruction quality and causing both FID curves to rise. In contrast, smaller $\gamma$ values make $\mathcal{E}$ evolve more gradually, giving the model enough time to optimize with respect to the current set. This leads to more stable training, delays degradation, and achieves lower overall FIDs. Hence, in practice, a relatively small $\gamma$ is generally preferable.

**Identifiability vs. practical recoverability.** As discussed in Sec 3, although injective corruption operators in principle allow recovery of $p_{\text{data}}$ from corrupted samples alone, the unfavourable sample-complexity rates make this practically infeasible. Fig 3c shows the iteration-wise FID of classical SFBD-OMNI under additive Gaussian noise with no clean samples ($\lambda = 0$). The steadily decreasing FID is consistent with Prop 4, which states $D_{\text{KL}} \left( p_{\text{data}} \parallel p^k \right)$ decreases monotonically (as $h^\dagger = p_{\text{data}}$ if the corruption is injective and $\lambda = 0$), starting from the untrained model $p^0$. However, even after saturating around iteration 4, the FID remains at $80.37$–substantially worse than the $10.81$ achieved when just $50$ clean samples are provided. This gap supports our claim: relying solely on corrupted samples is impractical, whereas even a very small clean set dramatically alleviates the issue.

**Further empirical evaluation and insights on practical limitations.** To further demonstrate the effectiveness of SFBD-OMNI, we evaluate it on high-resolution satellite and MRI images with Poisson and compressive sensing corruption; see Appx K. The results support our theoretical findings, yet the remaining artifacts suggest that achieving deployment-quality reconstructions may require domain-aware priors or problem-specific design choices.

## 7 DISCCUSION

In this work, we proposed SFBD-OMNI, a principled framework for distribution recovery based on diffusion-related models. Unlike GAN-based approaches that rely on adversarial training, our method builds on the Donsker–Varadhan representation of the KL divergence, which reveals an equivalence to a one-sided entropic optimal transport objective. This reformulation naturally yields an alternating minimization scheme that is theoretically grounded and practically stable.

Our analysis shows that SFBD-OMNI can recover the clean data distribution under identifiability conditions, and with the aid of a small set of clean samples, it remains effective even when these conditions fail. To address the computational challenges of sequential training, we introduced an online variant that enables end-to-end optimization without sacrificing optimality guarantees. Experiments on CIFAR-10 and CelebA confirm that the proposed method achieves consistent improvements over representative baselines across a range of corruption processes.

ACKNOWLEDGMENTS

We gratefully acknowledge funding support from NSERC, the Canada CIFAR AI Chairs program and the Ontario Early Researcher program. Resources used in preparing this research were provided, in part, by the Province of Ontario, the Government of Canada through CIFAR, and companies sponsoring the Vector Institute. We also thank Giannis Daras for his valuable insights and advice.

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

## A   LLM USAGE

LLMs were used to assist with text refinement and data formatting, but not for generating core research content.

## B   RELATED WORK

Recovering the underlying clean distribution from noisy or incomplete observations has been an active line of research in recent years. Bora et al. (2018) introduced AmbientGAN, demonstrating both theoretically and empirically that GANs can recover the true distribution even when only corrupted samples, such as randomly masked images, are available. Extending this idea, Wang et al. (2023) showed that, under mild assumptions, if corrupted real and generated samples are indistinguishable, the learned model necessarily recovers the ground-truth distribution. In this work, we present a comprehensive study of the identifiability conditions of corruption processes. When these conditions are satisfied, the clean distribution can be recovered directly from corrupted data. When they are not, we propose an effective strategy that leverages a small number of clean samples to enable substantial recovery of the underlying distribution.

Building on the success of training GANs with corrupted data, several recent studies have investigated whether diffusion models can also be trained under additive Gaussian corruption (Daras & Dimakis, 2023; Daras et al., 2024). Daras et al. (2024) demonstrated that when corruption is induced by a forward diffusion process, the marginal distribution at any time step constrains those at all other steps through a set of consistency relations. Exploiting this property, they showed that training on distributions above the corruption noise level allows the model to infer distributions at lower noise levels by enforcing consistency—an approach that has proven effective for fine-tuning latent diffusion models. Nevertheless, subsequent work found that training such models from scratch is impractical, as it would require an unrealistically large number of corrupted samples (Lu et al., 2025; Daras et al., 2025a). To address this, both Lu et al. (2025; 2026) and Daras et al. (2025a) proposed augmenting training with a small set of copyright-free clean samples, demonstrating that diffusion models can indeed be trained from scratch to achieve strong performance, albeit through distinct methodological routes.

Beyond recovering data distributions from corrupted observations, there has been growing interest in leveraging pretrained diffusion models to solve inverse problems, where the goal is to reconstruct underlying images from corrupted inputs (Chung et al., 2023; Feng et al., 2023; Zhang et al., 2023; Chung et al., 2022; Song et al., 2022; Murata et al., 2023). While these methods also perform recovery, they operate in a fundamentally different regime: they assume access to a pretrained diffusion model that already encodes the ground-truth distribution. By contrast, our work addresses the from-scratch setting, where the objective is to learn the ground-truth distribution itself directly from corrupted samples, without relying on a pretrained model.

Very recently, Daras et al. (2025b) proposed a complementary strategy, Ambient-o, which also seeks to align corrupted samples with the clean distribution. The method adds extra Gaussian noise to corrupted inputs so that, once sufficiently noised, they become nearly indistinguishable from clean–noisy samples and can be used directly for diffusion-model training – an idea similar in spirit to SDEdit (Meng et al., 2022). Because the alignment is achieved through noising rather than modelling the corruption, Ambient-o is agnostic to the underlying corruption process and requires no knowledge of it. However, this design introduces a trade-off: stronger noising improves distributional alignment but may also remove informative structure from the observations. In contrast, by assuming access to the corruption process as a black-box generator, SFBD-OMNI requires no extra noise and therefore preserves the full signal.

## C   GAUSSIAN NOISE REGULARIZATION FOR DETERMINISTIC SAMPLING PATHS

In Sec 2, we discuss how bridge models can be employed to learn the posterior distribution

$$u_{\mathbf{y}}(\mathbf{x}) = \frac{\mu(\mathbf{x})\, r(\mathbf{y} \mid \mathbf{x})}{\int \mu(\mathbf{x}')\, r(\mathbf{y} \mid \mathbf{x}')\mathrm{d}\mathbf{x}'},$$

given access to samples from $\mu(\mathbf{x})$ and the ability to query the corruption kernel $r(\mathbf{y} \mid \mathbf{x})$ as a black-box generator.

A challenge arises when the interpolation path is chosen to be the straight-line segment between $\mathbf{x}$ and $\mathbf{y}$. In this case, the backward sampling scheme in Eq (8) reduces to the special case $g = 0$, so sampling is performed by solving an ODE. Because the dynamics is deterministic, the model can no longer represent a distribution, leading to a degeneracy.

To avoid this issue, we perturb $\mathbf{y}$ with a small Gaussian noise before using it as the endpoint $\mathbf{x}_1$ in both training and sampling. This restores the stochasticity of the interpolation: the model learns a deterministic flow transporting $u_{\mathbf{y}}(\mathbf{x})$ to $\mathcal{N}(\mathbf{y}, \sigma^2 \mathbf{I})$, which aligns well with standard flow-matching formulations (Lipman et al., 2023; Liu et al., 2022).

Importantly, the perturbation is applied only to the endpoint used as the ODE's initial condition; the model itself is still conditioned on the original, unperturbed $\mathbf{y}$. Thus, the perturbation alters only the sampling path, not the conditioning variable.

From a conditional VAE perspective, this is similar to adding a small noise to the latent code $\mathbf{z}$ while keeping the conditioning variable fixed (e.g., a class label or observed image). Such perturbations regularize the decoder but do not change the underlying conditional distribution $p(\mathbf{x} \mid \mathbf{y})$ being modelled. Likewise, in our setting, the flow model continues to learn the correct posterior $u_{\boldsymbol{\theta}}(\mathbf{x} \mid \mathbf{y})$ because $\mathbf{y}$, the variable that defines the conditional law, remains unchanged. The perturbation merely prevents degeneracy in the ODE initialization and does not distort the learned conditional mapping.

## D  THEORETICAL RESULTS RELATED TO THE IDENTIFIABILITY

**Proposition 1** (Identifiability Condition). *Let $\mathcal{P}(X)$ denote the set of clean sample distributions. When the corruption kernel $r(\cdot \mid \mathbf{x})$ depends continuously on $\mathbf{x}$, the convex objective in Eq (10) admits a unique minimizer $p^* = p_{data}$ whenever $\mathcal{T}_r$ is injective on $\mathcal{P}(X)$. If $\mathcal{T}_r$ is not injective, the objective is still convex, but all distributions $p$ satisfying $\mathcal{T}_r p = \mathcal{T}_r p_{data}$ are minimizers.*

*Proof.* Let $q := \mathcal{T}_r p_{\mathrm{data}}$ and define

$$J(p) := D_{\mathrm{KL}}\left(q \parallel \mathcal{T}_r p\right).$$

*Convexity.* For $p_1, p_2 \in \mathcal{P}(X)$ and $t \in (0, 1)$,

$$\mathcal{T}_r\big(t p_1 + (1-t) p_2\big) = t\, \mathcal{T}_r p_1 + (1-t)\, \mathcal{T}_r p_2.$$

Since the map $m \mapsto D_{\mathrm{KL}}\left(q \parallel m\right)$ is strictly convex,

$$J\big(t p_1 + (1-t) p_2\big) = D_{\mathrm{KL}}\left(q \parallel t \mathcal{T}_r p_1 + (1-t) \mathcal{T}_r p_2\right) < t\, J(p_1) + (1-t)\, J(p_2).$$

*Injective case.* Assume $\mathcal{T}_r$ is injective on $\mathcal{P}(X)$. If $p_1 \neq p_2$ then $\mathcal{T}_r p_1 \neq \mathcal{T}_r p_2$, and by *strict* convexity of $m \mapsto D_{\mathrm{KL}}\left(m \parallel q\right)$,

$$J\big(t p_1 + (1-t) p_2\big) < t\, J(p_1) + (1-t)\, J(p_2) \qquad (t \in (0, 1)).$$

Thus $J$ is strictly convex in $p$. Since $J(p_{\mathrm{data}}) = D_{\mathrm{KL}}\left(q \parallel q\right) = 0$, $p_{\mathrm{data}}$ is the unique minimizer, i.e., $p^* = p_{\mathrm{data}}$. (Continuity of $\mathbf{x} \mapsto \int f(\mathbf{y})\, r(\mathbf{y} \mid \mathbf{x}) \mathrm{d}\mathbf{y}$ for bounded continuous $f$ gives the usual l.s.c./compactness to ensure well-posedness; uniqueness comes from strict convexity.)

*Non-injective case.* If $\mathcal{T}_r$ is not injective, then for any $p$ with $\mathcal{T}_r p = q$,

$$J(p) = D_{\mathrm{KL}}\left(q \parallel q\right) = 0,$$

which is the global minimum. Hence every $p \in \mathcal{S}(q) := \{p \in \mathcal{P}(X) : \mathcal{T}_r p = q\}$ is a minimizer.  $\square$

**Proposition 2.** *Let $h^\dagger = \arg\min_{p \in \mathcal{S}(q)} D_{\mathrm{KL}}\left(h \parallel p\right)$ denote the Information-projection of $h$ onto the original KLAP solution set. Then the minimizer of Eq (11), $p_\lambda^*$, converges to $h^\dagger$ as $\lambda \to 0$.*

*Proof.* Let $\mathcal{T}_r p = m_p$ and define

$$F(p) := D_{\mathrm{KL}}(q\|\mathcal{T}_r p), \qquad G(p) := D_{\mathrm{KL}}(h\|p).$$

For $\lambda > 0$ define

$$p_\lambda^\star \in \arg\min_p \Big\{ F(p) + \lambda\, G(p) \Big\},$$

Then the optimality against $h^\dagger$ gives, for each $\lambda > 0$,

$$F(p_\lambda^\star) + \lambda\, G(p_\lambda^\star) \;\le\; F(h^\dagger) + \lambda\, G(h^\dagger) \;=\; \lambda\, G(h^\dagger),$$

since $F(h^\dagger) = 0$. Hence

$$0 \;\le\; F(p_\lambda^\star) \;\le\; \lambda\big(G(h^\dagger) - G(h_\lambda^\star)\big) \;\le\; \lambda G(h^\dagger) \;\to\; 0.$$

By compactness of $\mathcal{P}(X)$, pick a subsequence $\lambda_k \downarrow 0$ with $p_{\lambda_k}^\star \to \bar{p}$. By lower semicontinuity of $F$, $F(\bar{p}) \le \liminf_k F(p_{\lambda_k}^\star) = 0$, hence $\bar{p} \in \mathcal{S}(q)$. From the same inequality, $\lambda_k G(p_{\lambda_k}^\star) \le \lambda_k G(h^\dagger)$, so dividing by $\lambda_k > 0$ and taking $\limsup$ gives $\limsup_k G(p_{\lambda_k}^\star) \le G(h^\dagger)$. By lower semicontinuity of $G$ and convergence $p_{\lambda_k}^\star \to \bar{p}$,

$$G(\bar{p}) \;\le\; \liminf_k G(p_{\lambda_k}^\star) \;\le\; \limsup_k G(p_{\lambda_k}^\star) \;\le\; G(h^\dagger).$$

Thus $G(\bar{p}) = \min_{p\in\mathcal{S}(q)} G(p)$. As the minimizer $h^\dagger$ is unique, $p_\lambda^\star \to h^\dagger$. $\qquad\square$

## E  THEORETICAL RESULTS RELATED TO THE ONE-SIDED OT

**The derivation of Eq (15).**

$$\begin{aligned}
D_{\mathrm{KL}}\left(q \;\|\; \mathcal{T}_p\right) &= \int q(\mathbf{y}) \log \frac{q(\mathbf{y})}{\mathcal{T}_r p(\mathbf{y})}\mathrm{d}\mathbf{y} = \int q(\mathbf{y})\log \frac{q(\mathbf{y})}{\int p(\mathbf{x}')r(\mathbf{y}\mid\mathbf{x}')\mathrm{d}\mathbf{x}'}\mathrm{d}\mathbf{y} \\
&= -\int q(\mathbf{y})\log \int p(\mathbf{x}')\,r(y\mid\mathbf{x}')\mathrm{d}\mathbf{x}' + C \\
&= -\int q(\mathbf{y})\log \mathbb{E}_p\big(\exp f_\mathbf{y}(\mathbf{x})\big) + C \\
&= -\int q(\mathbf{y})\max_{u_\mathbf{y}}\Big[\mathbb{E}_{u_\mathbf{y}}[f_\mathbf{y}(\mathbf{x})] - D_{\mathrm{KL}}\left(u_\mathbf{y}\;\|\;p\right)\Big] + C \\
&= \min_{u_\mathbf{y}}\mathbb{E}_q\left[D_{\mathrm{KL}}\left(u_\mathbf{y}\;\|\;p\right) - \mathbb{E}_{u_\mathbf{y}}[f_\mathbf{y}(\mathbf{x})]\right] + C,
\end{aligned}$$

where we have applied Eq (14), the Donsker-Varadhan variational principle (Donsker & Varadhan, 1983) in the second last equation.

**Lemma 1.** *Given the cost function be $c(\mathbf{x},\mathbf{y}) = -\log r(\mathbf{y}\mid\mathbf{x})$ for some corruption kernel $r$, consider the problem*

$$\min_{\pi\in\Pi_\mathbf{y}(q)} \iint \pi(\mathbf{x},\mathbf{y})\,c(\mathbf{x},\mathbf{y})\,d\mathbf{x}\,d\mathbf{y} + D_{\mathrm{KL}}\big(\pi\;\|\;p\otimes q\big),$$

*where $\Pi_\mathbf{y}(q)$ is the set of joint distributions with fixed $\mathbf{y}$-marginal $q$. If $q$ is realizable under $p$ via $r$, i.e. $p \in \mathcal{S}(q)$, then the optimizer is*

$$\pi^\star(\mathbf{x},\mathbf{y}) = p(\mathbf{x}\mid\mathbf{y})\,q(\mathbf{y}),$$

*which has marginals $\pi_\mathbf{x}^\star = p$ and $\pi_\mathbf{y}^\star = q$.*

*Proof.* Introducing a Lagrange multiplier for the constraint $\int \pi(\mathbf{x},\mathbf{y})\,d\mathbf{x} = q(\mathbf{y})$, the optimal solution takes the form

$$\pi^\star(\mathbf{x},\mathbf{y}) = \frac{p(\mathbf{x})\,q(\mathbf{y})\,e^{-c(\mathbf{x},\mathbf{y})}}{Z(\mathbf{y})}, \qquad Z(\mathbf{y}) = \int p(\mathbf{x})\,e^{-c(\mathbf{x},\mathbf{y})}\,d\mathbf{x}.$$

With $c(\mathbf{x}, \mathbf{y}) = -\log r(\mathbf{y} \mid \mathbf{x})$, this becomes

$$\pi^\star(\mathbf{x} \mid \mathbf{y}) \propto p(\mathbf{x}) \, r(\mathbf{y} \mid \mathbf{x}).$$

If $q(\mathbf{y}) = \int p(\mathbf{x}) \, r(\mathbf{y} \mid \mathbf{x}) \, d\mathbf{x}$, then $Z(\mathbf{y}) = q(\mathbf{y})$, and thus

$$\pi^\star(\mathbf{x} \mid \mathbf{y}) = \frac{p(x) r(\mathbf{y} \mid \mathbf{x})}{q(\mathbf{y})} = p(\mathbf{x} \mid \mathbf{y}).$$

Therefore,

$$\pi^\star(\mathbf{x}, \mathbf{y}) = q(\mathbf{y}) \, p(\mathbf{x} \mid \mathbf{y}),$$

which indeed has marginals $\pi_X^\star = p$ and $\pi_Y^\star = q$. $\qquad\square$

**Proposition 3.** *Define the cost function $c(\mathbf{x}, \mathbf{y}) := -\log r(\mathbf{y} \mid \mathbf{x})$. Problem (16) is equivalent to*

$$\arg\min_p \Phi(p) + \lambda \, D_{\mathrm{KL}}\left(h \parallel p\right)$$

*with*

$$\Phi(p) := \min_{\pi \in \Pi_{\mathbf{y}}(q)} \iint c(\mathbf{x}, \mathbf{y}) \, \pi(\mathbf{x}, \mathbf{y}) \, \mathrm{d}\mathbf{x}\mathrm{d}\mathbf{y} + D_{\mathrm{KL}}\left(\pi \parallel p \otimes q\right)$$

*where $\Pi_{\mathbf{y}}(q)$ denotes the set of joint distributions of $(\mathbf{x}, \mathbf{y})$ with $\mathbf{y}$-marginal fixed to $q$. Moreover, when $\lambda = 0$, the optimal solution $p^*$ coincides with the $\mathbf{x}$-marginal of the corresponding minimizer $\pi^*$ in the inner problem.*

*Proof.* We note that, by definition, $c = -f_{\mathbf{y}}$. Then starting from Eq (15), we have

$$\min_{u_{\mathbf{y}}} \mathbb{E}_q \left[ D_{\mathrm{KL}}\left(u_{\mathbf{y}} \parallel p\right) - \mathbb{E}_{u_{\mathbf{y}}}[f_{\mathbf{y}}(\mathbf{x})] \right]$$

$$= \min_\pi \iint \pi(\mathbf{x}, \mathbf{y}) \log \frac{\pi(\mathbf{x}, \mathbf{y})}{p(\mathbf{x}) q(\mathbf{y})} \mathrm{d}\mathbf{x}\mathrm{d}\mathbf{y} + \iint \pi(\mathbf{x}, \mathbf{y}) c(\mathbf{x}, \mathbf{y}) \mathrm{d}\mathbf{x}\mathrm{d}\mathbf{y}$$

$$= \min_\pi \iint \pi(\mathbf{x}, \mathbf{y}) c(\mathbf{x}, \mathbf{y}) \mathrm{d}\mathbf{x}\mathrm{d}\mathbf{y} + D_{\mathrm{KL}}\left(\pi \parallel p \otimes q\right).$$

where $\pi(\mathbf{x}, \mathbf{y}) = u_{\mathbf{y}}(\mathbf{x}) \, q(\mathbf{y})$ consisting of all joint distributions of $\mathbf{x}$ and $\mathbf{y}$ with the $\mathbf{y}$-marginal equal to $q$.

For the second part of the proposition, when $\lambda = 0$, our discussion in Sec 3.2 shows we have $p^* \in \mathcal{S}(q)$. Therefore, by Lem 1, we complete the proof. $\qquad\square$

## F  THEORETICAL RESULTS RELATED TO SFBD-OMNI

**Proposition 4** (Convergence to the optimum). *Let the distribution sequences $\{u_y^k\}$ and $\{p^k\}$ evolve according to Eq (18), with $\tilde{p}^k$ updated by Eq (19). Starting from an arbitrary initialization $p^0$ and for $\gamma \in (0, 1]$, under mild assumptions, we have $p^k \to p_\lambda^*$ as $k \to \infty$. Moreover, when $\lambda \to 0$, we have*

$$\lim_{k \to \infty} p^k = h^\dagger, \quad D_{\mathrm{KL}}\left(h^\dagger \parallel p^{k+1}\right) \leq D_{\mathrm{KL}}\left(h^\dagger \parallel p^k\right). \tag{20}$$

*In addition, the following bounds hold:*

$$\min_{1 \leq k \leq K} D_{\mathrm{KL}}\left(q \parallel \mathcal{T}_r p^k\right) \leq \frac{D_{\mathrm{KL}}\left(h^\dagger \parallel p^0\right)}{\gamma K}, \tag{21}$$

*where $K$ denotes the total number of iterations and $q = \mathcal{T}_r p_{data}$.*

**Convergence to $p_\lambda^*$.** By collecting the terms involving $u_{\mathbf{y}}$, $\mathcal{F}_\lambda(p^k, u_{\mathbf{y}})$ defined in Eq (16) can be written as

$$\mathcal{F}_\lambda(p^k, u_{\mathbf{y}}) = \mathbb{E}_q\left(D_{\mathrm{KL}}\left(u_{\mathbf{y}} \parallel u_{\mathbf{y}}^k\right)\right) + A_k \tag{22}$$

where $u_{\mathbf{y}}^k(x)$ is defined in Eq (18) and $A_k$ contains all the terms independent of $u_{\mathbf{y}}$. As a result, taking the minimizer of the objective in Eq (22) gives the update rule of $u_{\mathbf{y}}$ in Eq (18). Note that the result also shows that:

$$\mathcal{F}_\lambda(p^k, u_{\mathbf{y}}^{k-1}) - \mathcal{F}_\lambda(p^k, u_{\mathbf{y}}^k) = \mathbb{E}_q \left[ D_{\mathrm{KL}} \left( u_{\mathbf{y}} \parallel u_{\mathbf{y}}^k \right) \right]. \tag{23}$$

In addition, according to the Donsker-Varadhan variational principle (Donsker & Varadhan, 1983), when $u_{\mathbf{y}}$ is picked to the minimizer of the current $p^k$, we have

$$\mathcal{F}_{\mathbf{y}}(p^k, u_{\mathbf{y}}^{k+1}) = \mathcal{J}_\lambda(p_k), \tag{24}$$

with $\mathcal{J}_\lambda(p_k)$ defined in Eq (11).

When $\tilde{p}^k$ is updated in an incrementable way as shown in Eq (19), we claim $p^{k+1}$ is updated by minimizing

$$\mathcal{F}_\lambda(p, u_{\mathbf{y}}^k) + \nu D_{\mathrm{KL}} \left( p^k \parallel p \right) \tag{25}$$

with $\nu = \frac{(1-\gamma)(1+\lambda)}{\gamma}$. Notably, when updating ratio $\gamma = 1$, the entire sampling set $\mathcal{E}$ will be replaced, and we recover the original SFBD-OMNI. In this case, $\nu = 0$, the update of $p^k$ is then obtained by taking the minimizer of $\mathcal{F}_\lambda(p, u_{\mathbf{y}}^k)$ with $u_{\mathbf{y}}^k$ fixed.

Note that

$$\mathcal{F}_\lambda(p, u_{\mathbf{y}}^k) + \nu D_{\mathrm{KL}} \left( p^k \parallel p \right) = (1 + \lambda + \nu) D_{\mathrm{KL}} \left( \frac{1}{1+\lambda+\nu}(m_p^k + \lambda h + \nu p^k) \parallel p \right) + B_k, \tag{26}$$

where $B_k$ collects all the terms not involving $p$ and

$$m_p^k(\mathbf{x}) = \int q(\mathbf{y}) u_{\mathbf{y}}^k(\mathbf{x}) \mathrm{d}\mathbf{y}. \tag{27}$$

If we take $p^{k+1}$ as the minimizer of Eq (26), we have

$$p^{k+1} = \frac{1}{1+\lambda+\nu}(m_p^k + \lambda h + \nu p^k). \tag{28}$$

By choosing $\nu = \frac{(1-\gamma)(1+\lambda)}{\gamma}$, the update rule of $p^{k+1}$ coincides with the one in Eq (18) with $\tilde{p}^k$ updated according to Eq (19).

To see this, we note that, when $\nu = \frac{(1-\gamma)(1+\lambda)}{\gamma}$, the weights of $m_p^k = \int q(\mathbf{y}) u_{\mathbf{y}}^k(\mathbf{x}) \mathrm{d}\mathbf{y}$ in Eq (18) and Eq (28) are matched and equal to $\frac{\gamma}{1+\lambda}$. In addition, the weight ratios between $m_p^{k-1}$ and $m_p^k$ (absorbed respectively in $p^k$ in Eq (28) and $\tilde{p}^k$ in Eq (19)) are both $1 - \gamma$. This suggests that for both update rules, $m_p^k$'s are mixed in exactly the same way. As a result, the two update rules must be equivalent.

The optimiality of $p^{k+1}$ also suggest,

$$\left( \mathcal{F}_\lambda(p^k, u_{\mathbf{y}}^k) + \nu D_{\mathrm{KL}} \left( p^k \parallel p^k \right) \right) - \left( \mathcal{F}_\lambda(p, u_{\mathbf{y}}^k) + \nu D_{\mathrm{KL}} \left( p^k \parallel p^{k+1} \right) \right)$$
$$= (1 + \lambda + \nu) D_{\mathrm{KL}} \left( \frac{1}{1+\lambda+\nu}(m_p^k + \lambda h + \nu p^k) \parallel p^k \right),$$

which implies

$$\mathcal{F}_\lambda(p^k, u_{\mathbf{y}}^k) - \mathcal{F}_\lambda(p^{k+1}, u_{\mathbf{y}}^k) \geq 0. \tag{29}$$

As a result, according to Eq (23) and Eq (29), we have $\mathcal{F}_\lambda(p^k, u_{\mathbf{y}}^k) \rightarrow \mathcal{F}_\lambda(p^k, u_{\mathbf{y}}^{k+1}) \rightarrow \mathcal{F}_\lambda(p^{k+1}, u_{\mathbf{y}}^{k+1})$ decreases monotonically. Combined with Eq (24), we have $\mathcal{J}_\lambda(p_k)$ decreases monotonically as well. In addition, as $\mathcal{J}_\lambda(p_k)$ is bounded below, $\mathcal{J}_\lambda(p_k)$ much converge to some limit $\mathcal{J}_\lambda^\infty$. As a result, for every subsequence of $p_k$, it must converge to some cluster point $\bar{p}$, where $\bar{p}$ is then a fixed point under the update rules in Eq (18). That is,

$$\bar{p} = \frac{1}{1+\lambda+\nu}(m_{\bar{p}} + \lambda h + \nu \bar{p}),$$

where $m_{\bar{p}}(\mathbf{x}) = \int q(\mathbf{y})\bar{u}_{\mathbf{y}}(\mathbf{x})\mathrm{d}\mathbf{y}$ and the posterier distribution $\bar{u}_{\mathbf{y}}(\mathbf{x}) = \bar{p}(\mathbf{x} \mid \mathbf{y}) = \frac{\bar{p}(\mathbf{x})r(\mathbf{x}|\mathbf{y})}{\mathcal{T}_r\bar{p}(\mathbf{y})}$. After rearrangement, we have

$$\frac{1}{1+\lambda}m_{\bar{p}} + \frac{\lambda}{1+\lambda}h = \bar{p}. \tag{30}$$

We complete the proof of the optimal convergence by showing that the only $p$ satisfying Eq (30) is $p_\lambda^*$.

**Lemma 2.** *The following are equivalent for such $p$:*

$$p(\mathbf{x}) = \frac{1}{1+\lambda} \int q(\mathbf{y})\, p(\mathbf{x}|\mathbf{y})\,\mathrm{d}y + \frac{\lambda}{1+\lambda}\, h(\mathbf{x}), \tag{31}$$

$$\exists\, \mu \in \mathbb{R} \ \ s.t. \ -\int \frac{q(\mathbf{y})}{\mathcal{T}_r(\mathbf{y})}\, r(\mathbf{y}|\mathbf{x})\,\mathrm{d}\mathbf{y} - \lambda\frac{h(\mathbf{x})}{p(\mathbf{x})} + \mu = 0 \tag{32}$$

*Moreover as $\mathcal{J}_\lambda$ is strictly convex on the probability simplex, so any solution of (31) is the unique global minimizer of $\mathcal{J}_\lambda$.*

*Proof.* $(\Rightarrow)$. Multiply (32) by $p(\mathbf{z})$ and apply

$$\int \frac{q(\mathbf{y})}{\mathcal{T}_r(\mathbf{y})}\, r(\mathbf{y}|\mathbf{x})\,\mathrm{d}\mathbf{y} = \frac{1}{p(\mathbf{x})}\int q(\mathbf{y})\, p(\mathbf{x}|\mathbf{y})\,\mathrm{d}\mathbf{y},$$

to obtain

$$\int q(\mathbf{y})\, p(\mathbf{x}|\mathbf{y})\,\mathrm{d}\mathbf{y} + \lambda\, h(\mathbf{x}) = \mu\, p(\mathbf{x}).$$

Integrate both sides over $\mathbf{x}$:

$$1 + \lambda = \mu \int p(\mathbf{x})\,\mathrm{d}\mathbf{x} = \mu \ \Rightarrow \ \mu = 1 + \lambda,$$

and substitute back to get (31).

$(\Leftarrow)$. Starting from (31), rearrange:

$$(1+\lambda)p(\mathbf{x}) - \lambda h(\mathbf{x}) = \int q(y)\, p(\mathbf{x}|\mathbf{y})\,\mathrm{d}\mathbf{y} = p(\mathbf{x})\int \frac{q(\mathbf{y})}{\mathcal{T}_r(\mathbf{y})}\, r(\mathbf{y}|\mathbf{x})\,\mathrm{d}\mathbf{y}.$$

Divide by $p(\mathbf{y}) > 0$ and rearrange:

$$-\int \frac{q(\mathbf{y})}{\mathcal{T}_r(\mathbf{y})}\, r(\mathbf{y}|\mathbf{x})\,\mathrm{d}\mathbf{y} - \lambda\frac{h(\mathbf{x})}{p(\mathbf{x})} + (1+\lambda) = 0,$$

which is (32) with $\mu = 1 + \lambda$.

In addition, the Lagrangian for $\min_{p\geq 0, \int p=1} \mathcal{J}_\lambda(p)$ is $\mathcal{L}(p) = \mathcal{J}_\lambda(p) + \mu\left(\int p - 1\right)$. For interior $p > 0$, the Gateaux derivative of $\mathcal{J}_\lambda$ at $p$ equals the left side of (32) minus $\mu$. Thus (32) is the first-order condition $\nabla\mathcal{L}(p) = 0$. Since $\mathcal{J}_\lambda$ is convex, any interior stationary point is a global minimizer. $\square$

By Lem 2, we know that $\bar{p} = p_\lambda^*$ is the unique minimizer of $\mathcal{J}_\lambda$ for $\lambda > 0$. Moreover, as $\lambda \to 0$, Prop 2 implies that $p_\lambda^* \to h^\dagger$, which establishes the first part of the statement.

**Convergence rate when $\lambda \to 0$.** Define $\mathcal{H}(p) = D_{\mathrm{KL}}\left(h^\dagger \parallel p\right)$. When $\lambda \to 0$, as $\nu = \frac{(1-\gamma)(1+\lambda)}{\gamma} = \frac{1-\gamma}{\gamma}$, Eq (28) reduces to

$$p^{k+1} = \gamma\, m_p^k + (1-\gamma)\, p^k, \tag{33}$$

where $m_p^k(\mathbf{x}) = \int q(\mathbf{y})u_{\mathbf{y}}^k(\mathbf{x})\mathrm{d}\mathbf{y}$ and $u_{\mathbf{y}}^k$ is updated according to Eq (18). Then by the convexity of the KL divergence, we have

$$\mathcal{H}(p^{k+1}) \leq (1-\gamma)\, \mathcal{H}(p^k) + \gamma\, \mathcal{H}(m_p^k). \tag{34}$$

Rearrangement yields

$$\mathcal{H}(m_p^k) \geq \mathcal{H}(p^k) + \frac{1}{\gamma}\left(\mathcal{H}(p^{k+1}) - \mathcal{H}(p^k)\right). \tag{35}$$

Let $H^\dagger$ denote the joint distribution induced by $h^\dagger(\mathbf{x})r(\mathbf{y}|\mathbf{x})$ and likewise $P^k$ the one by $p^k(\mathbf{x})r(\mathbf{y}|\mathbf{x})$. In addition, let $u_\mathbf{y}^\dagger$ denote the posterior distribution of $H^\dagger$. Note that, as $h^\dagger \in \mathcal{S}(q)$, we have $h^\dagger(\mathbf{x})r(\mathbf{y} \mid \mathbf{x}) = q(\mathbf{y})u_\mathbf{y}^\dagger(\mathbf{x})$. Then, by the disintegration theorem (Vargas et al., 2021), we have

$$\mathcal{H}(p^k) = D_{\mathrm{KL}}\left(h^\dagger \parallel p^k\right) = D_{\mathrm{KL}}\left(H^\dagger \parallel P^k\right) = D_{\mathrm{KL}}\left(q \parallel \mathcal{T}_r p^k\right) + \mathbb{E}_{H^\dagger}\left[D_{\mathrm{KL}}\left(u_\mathbf{y}^\dagger \parallel u_\mathbf{y}^k\right)\right]$$

$$\geq D_{\mathrm{KL}}\left(q \parallel \mathcal{T}_r p^k\right) + D_{\mathrm{KL}}\left(h^\dagger \parallel m_p^k\right) \overset{(35)}{\geq} D_{\mathrm{KL}}\left(q \parallel \mathcal{T}_r p^k\right) + \mathcal{H}(p^k) + \frac{1}{\gamma}\left(\mathcal{H}(p^{k+1}) - \mathcal{H}(p^k)\right).$$

Cancel out $\mathcal{H}(p^k)$ and rearrange to obtain the monotonic decrease of $D_{\mathrm{KL}}\left(h^\dagger \parallel p^k\right)$

$$-\gamma D_{\mathrm{KL}}\left(q \parallel \mathcal{T}p^k\right) \geq \mathcal{H}(p^{k+1}) - \mathcal{H}(p^k). \tag{36}$$

Telescoping it yields

$$\mathcal{H}(p^0) \geq \sum_{k=0}^{K}\left[\mathcal{H}(p^k) - \mathcal{H}(p^{k+1})\right] \geq \gamma \sum_{k=1}^{K} D_{\mathrm{KL}}\left(q \parallel \mathcal{T}p^k\right). \tag{37}$$

As a result,

$$\min_{k \in \{1,2,\dots,K\}} D_{\mathrm{KL}}\left(q \parallel \mathcal{T}p^k\right) \leq \frac{\mathcal{H}(p^0)}{\gamma K} = \frac{D_{\mathrm{KL}}\left(h^\dagger \parallel p^0\right)}{\gamma K}. \tag{38}$$

## G EXPERIMENT CONFIGURATIONS

All SFBD-OMNI models were trained on one to four L40 GPUs using a SLURM scheduling system. With the standard SFBD-OMNI, training on CIFAR-10 takes about 5 days and on CelebA about 8 days. The online variant is more efficient, requiring roughly 4 days for CIFAR-10 and 6 days for CelebA.

### G.1 MODEL ARCHITECTURES

We implement the proposed methods using the EDM backbone (Karras et al., 2022) without preconditioning, and adopt this configuration throughout our empirical studies. The training pipeline is built on flow matching (Lipman et al., 2023).

Table 2: Experimental Configuration for CIFAR-10 and CelebA

| Parameter | CIFAR-10 | CelebA |
|---|---|---|
| **General** | | |
| Batch Size | 256 | 256 |
| Loss Function | `Flow matching loss` (Lipman et al., 2023) | `Flow matching loss` (Lipman et al., 2023) |
| Denoising Method | torchdiffeq (Chen, 2018) | torchdiffeq (Chen, 2018) |
| Sampling Method | torchdiffeq (Chen, 2018) | torchdiffeq (Chen, 2018) |
| **Network Configuration** | | |
| Dropout | 0.3 | 0.3 |
| Channel Multipliers | $\{2, 2, 2\}$ | $\{2, 2, 2\}$ |
| Model Channels | 128 | 128 |
| Channel Mult Noise | 2 | 2 |
| **Optimizer Configuration** | | |
| Optimizer Class | `RAdam` (Kingma & Ba, 2015; Liu et al., 2020) | `RAdam` (Kingma & Ba, 2015; Liu et al., 2020) |
| Learning Rate | 0.0001 | 0.0001 |
| Betas | (0.9, 0.95) | (0.9, 0.95) |

## G.2 DATASETS

All experiments on CIFAR-10 (Krizhevsky & Hinton, 2009) and CelebA (Liu et al., 2015) are performed using only the training splits. For FID evaluation, each model generates 50,000 samples, and the score is computed against the entire training set.

# H SAMPLING RESULTS

## H.1 CIFAR-10

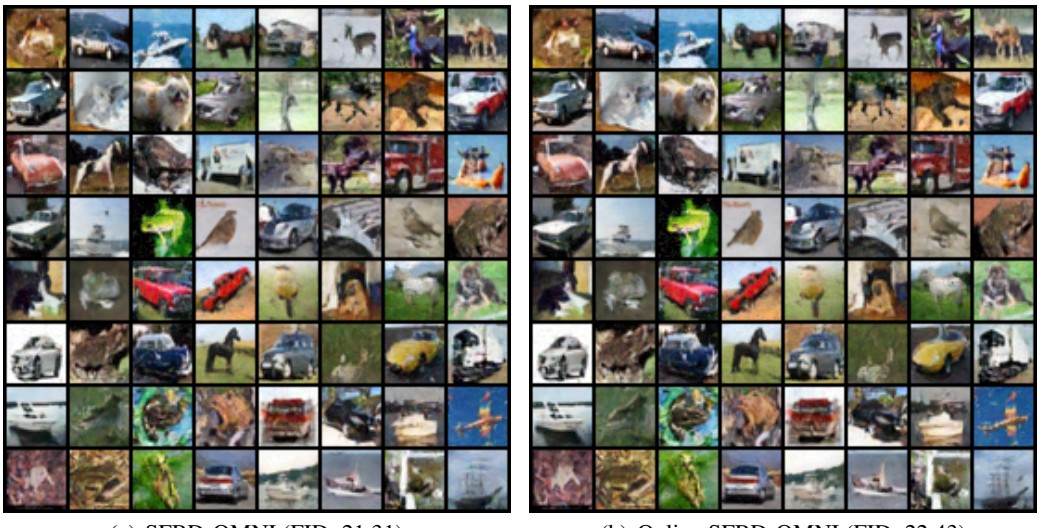

(a) SFBD-OMNI (FID: 21.31)    (b) Online SFBD-OMNI (FID: 22.43)

Figure 4: Pixel Masking

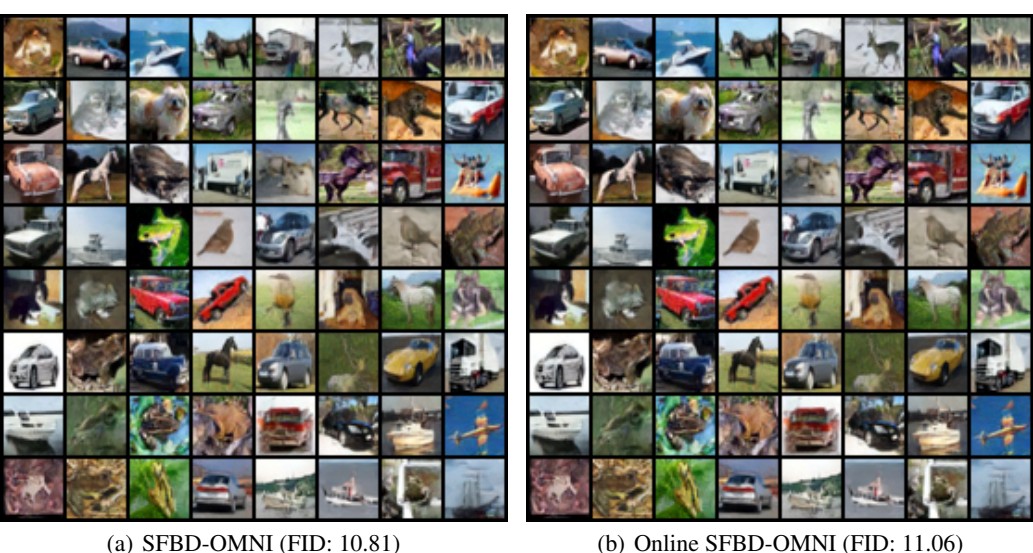

(a) SFBD-OMNI (FID: 10.81)    (b) Online SFBD-OMNI (FID: 11.06)

Figure 5: Addictive Gauss.

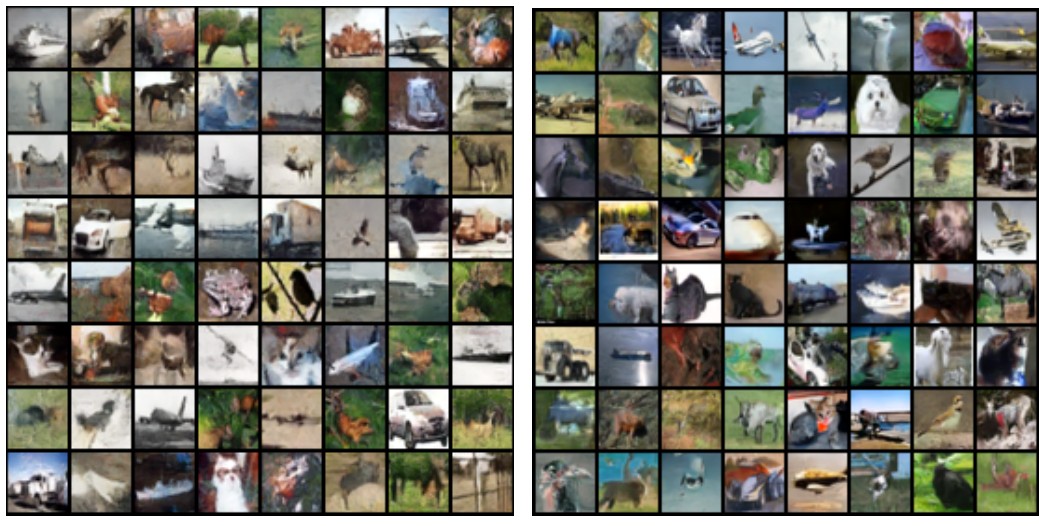

(a) SFBD-OMNI (FID: 32.61)          (b) Online SFBD-OMNI (FID: 31.32)

Figure 6: Grayscale

## H.2  CELEBA

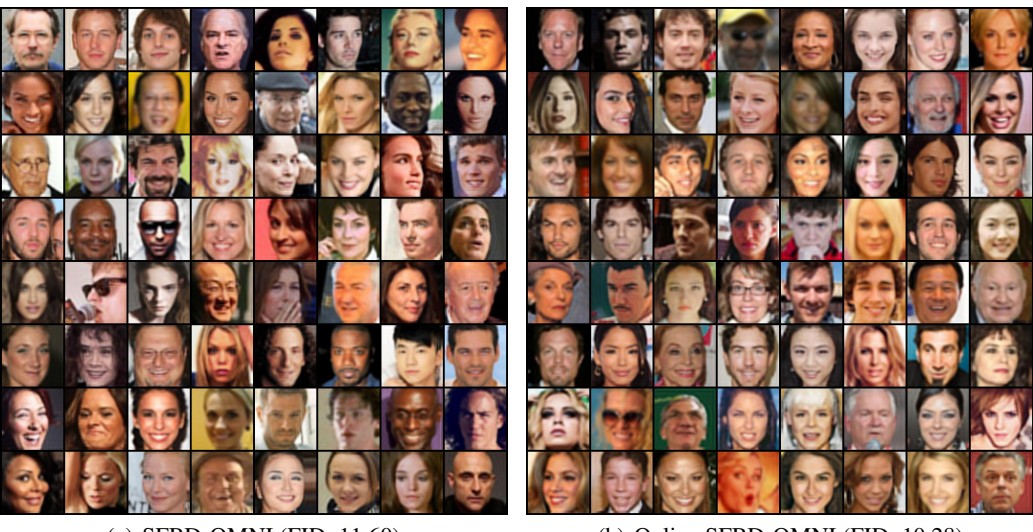

(a) SFBD-OMNI (FID: 11.60)          (b) Online SFBD-OMNI (FID: 10.28)

Figure 7: Gauss. Blur

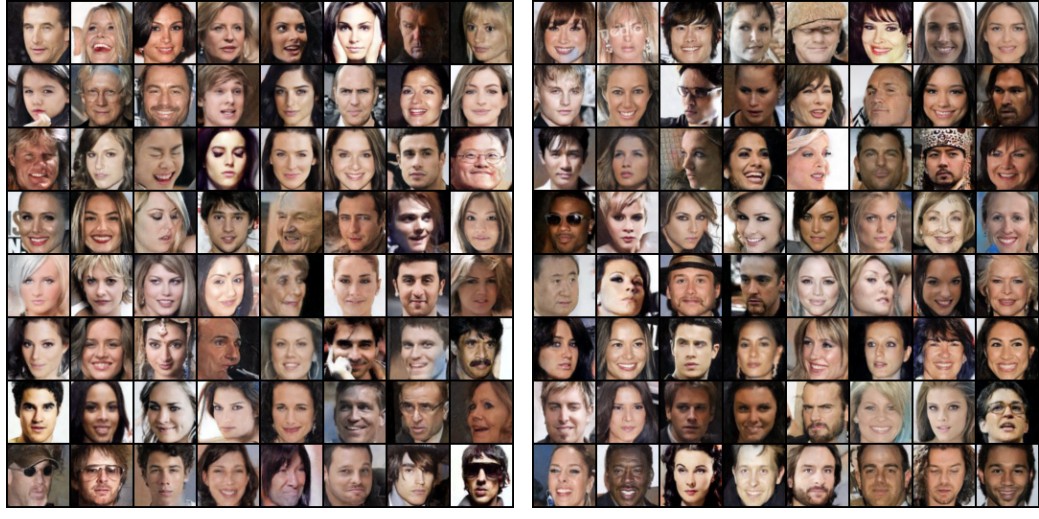

(a) SFBD-OMNI (FID: 11.85)      (b) Online SFBD-OMNI (FID: 11.21)

Figure 8: Grayscale

# I DISCUSSION ON AMBIENT DIFFUSION OMNI

Ambient diffusion-Omni (Ambient-o) incorporates corrupted samples by injecting additional Gaussian noise. The key idea is that once sufficient Gaussian noise is added, the corrupted-noisy distribution and the clean-noisy distribution become harder to distinguish. This observation suggests that a corrupted sample, after being further noised, can effectively be treated as a noised clean sample, allowing it to be used in standard diffusion-model training. This effect does not depend on the specific form of the corruption process, allowing AD-OMNI to operate without requiring knowledge of the corruption mechanism. However, this strategy comes with an inherent trade-off. While heavy noising helps align corrupted samples with clean ones, it also risks erasing useful structure and details within the observations. In other words, sufficient noise is needed for Ambient-o to function as intended, but excessive noise may suppress the informative signal that could otherwise benefit model learning.

In contrast, SFBD-OMNI does not inject additional noise into the samples and therefore preserves the full information of the observations. Rather than relying on excessive noising to align distributions, our method leverages knowledge of the corruption process itself, avoiding information destruction while still enabling effective training.

In Table 3, following the Ambient-o setting, we apply a Gaussian blur with varying strengths $\sigma$ and assume access to $10\%$ clean samples. The table shows that, by fully leveraging the information

| Blur Strength ($\sigma$) | Ambient-o | Online SFBD-OMNI |
|:---:|:---:|:---:|
| 0.6 | 5.34 | 0.97 |
| 1.0 | 6.16 | 3.07 |

Table 3: Ambient-o vs. online SFBD-OMNI: FID under Gaussian blur of varying strengths.

contained in the corrupted samples, SFBD-OMNI achieves substantially lower FID across blur levels, outperforming Ambient-o by a large margin.

# J PRETRAINING MODELS USING SAMPLES FROM A SIMILAR DISTRIBUTION

As mentioned in Sec 6, according to our theory, when the corruption function is identifiable, a small number of clean samples are needed only to obtain a good initial distribution $p_0$. This also implies

that if clean samples from the target distribution are unavailable, it is acceptable to use samples from a similar distribution instead. To demonstrate this, we pretrain the model on CIFAR-10 using clean samples from the truck class, and then apply iterative optimization to recover the distributions of automobile, ship, and horse, where all samples are corrupted by additive Gaussian noise with noise level $\sigma = 0.2$. The FID scores before and after iterative optimization are shown in Table 4.

| Class | After Pretrain | Final |
|---|---|---|
| Automobile | 8.36 | 6.19 |
| Ship | 13.96 | 8.78 |
| Horse | 25.87 | 13.55 |
| Horse (no pretrain) | – | 80.17 |

Table 4: FID comparison across CIFAR-10 classes before and after finetuning, with pretraining conducted on the truck class.

As the table shows, for classes similar to truck – such as automobile – the model successfully recovers the target distribution, as indicated by the low final FID. For classes that are less similar, pretraining still provides substantial benefits. In particular, for horse, pretraining on the truck class reduces the final FID dramatically from 80.17 (without pretraining) to 13.55, illustrating the importance of a good initial distribution even when the clean samples are drawn from a different – but related – class. (Notably, the horse and truck classes still share several low-level characteristics such as edges and common background elements like grass or road surfaces.)

## K  SUPPLEMENTARY EMPIRICAL RESULTS IN LATENT SPACE

In this section, we provide additional empirical results on high-resolution satellite (256 x 256) and MRI datasets (320 x 320) corrupted by Poisson noise and compressive sensing (CS). For MRI, the experiments are conducted in the latent space for computational efficiency. The results remain consistent with those in the main text, further validating the effectiveness of SFBD-OMNI across diverse corruption settings. At the same time, qualitative inspection reveals visible reconstruction artifacts, indicating remaining limitations and motivating future work that incorporates stronger priors or modality-specific inductive biases.

**Satellite images and Poisson noise.** We use satellite images from the training split of the NWPU-RESISC45 dataset (Cheng et al., 2017), which contains 45 scene classes with 600 images per class.

For this dataset, we consider Poisson noise corruption. Poisson noise arises naturally in photon-limited imaging systems, including satellite and remote-sensing cameras, where the number of detected photons per pixel is inherently stochastic and follows Poisson statistics (Hasinoff, 2014; Schott, 2007). Following common practice in Poisson-noise simulation studies, we vary the photon budget as $\alpha \in 10, 50, 100$, corresponding to severe, moderate, and mild shot-noise conditions, respectively. We simulate Poisson noise by interpreting each pixel intensity $x_{i,j,c} \in [0,1]$ as a normalized photon arrival rate and sampling

$$y_{i,j,c} \sim \frac{1}{\alpha} \operatorname{Poisson}(\alpha \, x_{i,j,c}),$$

followed by clipping the resulting values to the valid range $[0,1]$ (Makitalo & Foi, 2011).

**MRI image set and compressive sensing corruption.** We conduct our experiments on the fastMRI brain dataset (Zbontar et al., 2018), using its multicoil training subset, which provides fully sampled raw $k$-space data from clinical brain MRI scans. For each volume, we discard the final four slices, as these typically contain little or no brain anatomy. After filtering, the dataset contains 52,778 MRI slices, from which we randomly sample 2,000 as the clean set.

For this dataset, we consider the compressive sensing degradation, which is a natural corruption model for MRI. In particular, MRI scanners do not acquire images directly; instead, they measure the spatial frequencies of the underlying anatomy in $k$-space (Lustig et al., 2007). The acquisition process

| Stage | Satellite — Poisson Noise | | | MRI Compressive Sensing |
|---|---|---|---|---|
| | $\alpha = 10$ | $\alpha = 50$ | $\alpha = 100$ | |
| After Pretrain | 9.32 | 5.71 | 4.43 | 36.98 |
| Final Result | 7.11 | 4.13 | 3.40 | 28.71 |

Table 5: FID Scores of Online SFBD-OMNI for satellite images (Poisson noise with $\alpha = 10$, 50, 100) and MRI scans (compressive sensing).

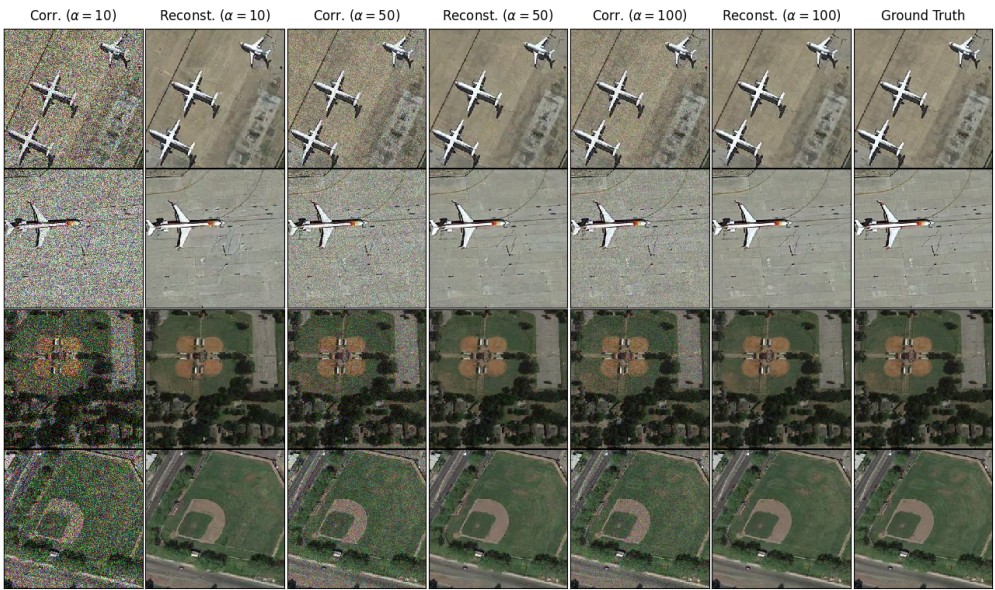

Figure 9: Reconstructed Satellite Images – Poisson Noise (photo budget $\alpha = 10, 50, 100$).

therefore corresponds to sampling the Fourier transform of the image. Because clinical MRI protocols routinely undersample $k$-space to shorten scan time, compressed-sensing MRI accelerates acquisition by collecting only a subset of frequency coefficients and relying on reconstruction algorithms to recover the missing data. Consequently, partial Fourier undersampling is not an artificial degradation, but a realistic and practically motivated corruption process for accelerated MRI.

To simulate a realistic compressive sensing degradation, we follow the standard undersampled MRI acquisition model of Lustig et al. (2007). Given an image $x \in \mathbb{R}^{H \times W}$, the corrupted observation is obtained by undersampling its Fourier transform:

$$y = P_\Omega(\mathcal{F}(x)) \,,$$

where $\mathcal{F}$ denotes the 2-D discrete Fourier transform and $P_\Omega$ is a binary mask selecting a subset $\Omega$ of frequency coefficients. We use a fixed variable-density sampling mask, generated once at the beginning of the experiment and reused for all samples. Following common practice in compressed-sensing MRI, the central low-frequency region of $k$-space (10% of the spatial extent) is fully sampled to preserve global structure, while coefficients outside this region are sampled independently with probability 0.20. This produces a realistic and reproducible compressive sensing corruption operator that retains essential low-frequency content while heavily undersampling high-frequency components.

**Implementation.** For satellite images, we continue using the model architectures described in Appx G. For the experiments on MRI, we use the pretrained autoencoder (VAE) from Stable Diffusion v1.5 (Rombach et al., 2022a) to encode images into the latent space and to decode the model outputs. We keep the model architectures described in Appx G unchanged, except for adjusting the input and output channels to 4 to match the dimensionality of the latent representations.

**Results.** Table 5 summarizes the FID performance of online SFBD-OMNI for satellite images with Poisson corruption and MRI scans under compressive sensing, measured after pretraining

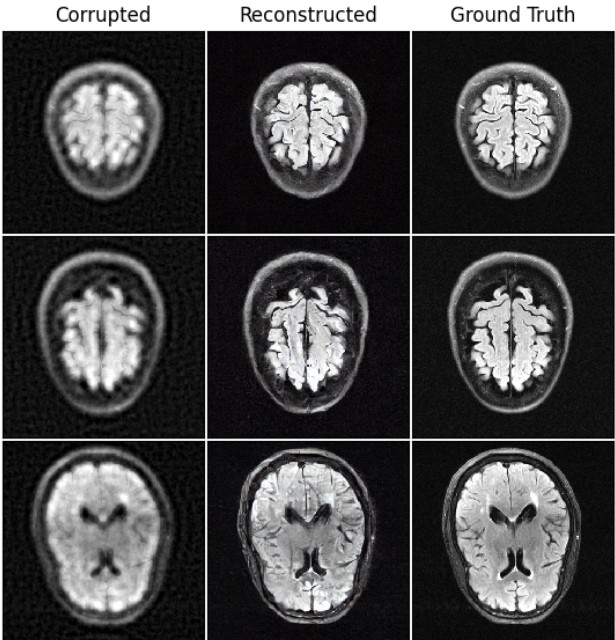

Figure 10: Reconstructed MRI – Compressive Sensing

and after online iterative refinement. Across all evaluated settings, the online phase yields a clear improvement over the pretrained model, demonstrating the effectiveness of SFBD-OMNI as a general reconstruction framework for real-world corruption processes.

We provide qualitative reconstructions in Figures 9 and 10. On satellite images, SFBD-OMNI visibly recovers large-scale structures – such as building layouts, runway geometry, and aircraft outlines – that are heavily disrupted by Poisson shot noise. In the MRI setting, despite severe undersampling, the method reconstructs coherent tissue boundaries and globally consistent anatomical structure.

**Limitations and noise sensitivity.** While our framework produces promising reconstructions, the Poisson-noise results also reveal a clear limitation. As the photon budget decreases (i.e., noise increases), output quality degrades, with more residual artifacts and reduced fine-grained fidelity. This trend is reflected quantitatively in Table 5, where performance drops moving from $\alpha=100$ to $\alpha=10$. Qualitative examples in Fig 9 further highlight these failure modes–under extreme shot noise, texture-level restoration remains challenging and fine structure is only partially recovered. Likewise, for MRI samples corrupted by compressive sensing, we still notice some visible artifacts as shown in Fig 10.

Across both Poisson-corrupted satellite imagery and compressive-sensed MRI scans, SFBD-OMNI reliably recovers the global structure of the underlying data distribution, but struggles to reconstruct fine details, particularly under severe corruption. Poisson noise reduces the amount of usable information in low-photon settings, and similarly, heavy MRI undersampling restricts the available signal for reconstruction. In such cases, recovering the clean distribution becomes inherently more challenging and may require stronger priors or model-specific inductive biases.

