# OpenReview forum: "SFBD-OMNI: Bridge models for lossy measurement restoration with limited clean samples"
_ICLR.cc/2026/Conference — ICLR 2026 Poster_

### Official Review · Reviewer_Sh1d · 2025-10-26

**Soundness:** 3
**Presentation:** 3
**Contribution:** 2
**Rating:** 6
**Confidence:** 4

**Summary:**

The paper introduces SFBD-OMNI, a new framework for generating from a clean data distribution while training from corrupted samples and a limited number of clean samples. The proposed method generalizes SFBD to arbitrary corruption models, providing both theoretical guarantees and practical algorithms, including an online variant that supports end-to-end training. The authors handle non-identifiable corruptions by introducing a regularization that acts as a "projection" (in the KL sense), enforcing the solution to be close to a given distribution constructed from a small set of clean samples.

**Strengths:**

1. Overall, the proposed approach is clean and elegant, relying on well-established principles.
2. The paper is well-written and easy to follow. The paper is structured in a clear and logical manner, and the authors provide several intuitive examples throughout to help the reader understand the concepts presented.
3. The experimental results seem good -- the proposed approach improves upon previous methods, and it is able to handle non-identifiable degradations such as de-colorization.

**Weaknesses:**

1. Only two datasets (CIFAR-10 and CelebA) are evaluated. Testing on more complex real-world domains (e.g., MRI, satellite data) could have been more appealing from a practical standpoint. In my opinion, this is a major weakness.
2. While clean-sample weighting is explored (Figure 2), the effect of other hyperparameters such as $\gamma$ (update ratio) remains unclear.
3. The authors perturb the endpoints $y$ to "prevent degeneration", since flow matching is a deterministic map. But an explanation about the effects of such perturbations (both in terms of theory and practice) is missing.
4. The proposed approach still uses a small number of clean samples to train the generative model, which is a limitation. In real-world scenarios and scientific applications (e.g., astrophysics), we often only have corrupted samples.

**Questions:**

Why was flow matching selected as the primary instantiation? Would other stochastic bridges (e.g., Schrodinger) behave differently? Perhaps using another bridge could alleviate the need to perturb the endpoint samples? It would be interesting to assess several types of flows, especially since, from the beginning of the paper, the authors formulate the problem in the stochastic sense (using Brownian motion, eq. 8), but then implement it with flow matching (which does not involve Brownian motion).

---

> ### Author Response · Authors · 2025-11-23
>
> We thank you for your constructive feedbacks. Our responses are as follows:
>
>
> **A1.** *Regarding the emprical results on real world domains.*
>
> **Q1.** Our choice of datasets follows established practice in the ambient-learning literature (e.g., SFBD). To address the the request for more realistic restoration scenarios, we added new experiments in Appendix I, including satellite images corrupted by Poisson noise and MRI data degraded by compressive-sensing operators. We also clarify why these corruption models are standard and well-motivated for their respective imaging modalities.
>
> Across these higher-resolution and more practical settings, the performance trends are consistent with our results on CIFAR-10 and CelebA. These additional experiments further corroborate our theoretical claims and highlight the robustness and practical utility of SFBD-OMNI.
>
> **Q2.** *Regarding the sensitivity study on hyperparameters*
>
> **A2.** In the revision, we added ablation studies on both the number of clean samples and the online update ratio $\gamma$ in Sec 6 (see Fig. 3). Combined with our existing analysis on clean-sample weighting, we hope these results now offer a more complete picture of the behavior of SFBD-OMNI.
>
> **Q3.** *Regarding the choice of the flow-matching implementation and other potential bridge implementations.*
>
> **A3.** In Sec 6 (Models and Other Configurations) and Appendix C, we have added further explanation and motivation for our choice of a flow-matching backbone. In brief, we select flow matching because it converges substantially faster than standard diffusion-based approaches. This efficiency is especially important in our framework, where the bridge models must be trained repeatedly against a moving target distribution. We have added additional theoretical justifications in Sec C to explain why purturbing $\mathbf{y}$ is sufficient to address the degeneration problem.
>
> Regarding alternative bridge parameterizations, we note that the formulation in (6) reduces to flow matching when $g = 0$. We also agree that a more systematic exploration of alternative bridge designs would be valuable and could further strengthen the framework. We view this as an important avenue that complements our current contribution and plan to investigate it in follow-up work.
>
>
> **Q4.** *Regarding uses a small number of clean samples to train the generative model.*
>
> **A4.** We acknowledge that requiring a small number of clean samples may limit the applicability of our method in certain settings. However, when the corruption process fails to satisfy the identifiability condition, our theory shows that recovering the original distribution is fundamentally impossible. Even when identifiability holds, the pessimistic sample-complexity results indicate that, without some clean samples, obtaining enough noisy observations to train a high-quality model is often practically infeasible. (Please also see the additional discussion on this point added at the end of Sec 3.2 in the revised manuscript.)
>
> That said, we agree that developing methods that do not rely on clean samples is both valuable and broadly useful. We view this as an important future direction, likely requiring additional structural assumptions to overcome the inherent sample-complexity and identifiability challenges.

---

> > ### Comment · Reviewer_Sh1d · 2025-11-23
> >
> > I thank the authors for their response.
> >
> > Q1: Thank you for your efforts in adding additional experiments. However, the results on satellite data with Poisson noise seem ineffective to me. I would be interested in seeing both ground-truth image examples and the corresponding generated samples, as the generated images seem to contain noticeable and undesirable artifacts. Related to this, the level of Poisson noise used in your experiments appears to be quite small. I suspect that increasing the noise level would amplify these artifacts.
> > Would it be possible for you to evaluate SFBD-OMNI under varying levels of Poisson noise? Additionally, it may be worthwhile to make the setting more realistic by incorporating geometric transformations (which commonly appear in satellite imagery). I do not believe this limitation is a reason to reject this paper, but I do think it is essential for the authors to clearly convey the practical limitations of the method, specifically by identifying practical scenarios where the method fails (such as the one suggested above). Currently, the paper mostly highlights the method’s successes, but this is partly because it is evaluated on relatively simple examples.
> >
> > Q2: Thank you.
> >
> > Q3: Thank you for improving your explanation of the choice of flow matching. It will indeed be valuable to include a systematic comparison of alternative bridge methods. Currently, I still view it as a limitation of the current paper.
> >
> > Q4: If clean samples are still required in practice, then what's the point of the theory in the paper, if it cannot be really used in practice? I thought that the idea is that if the corruption is identifiable, then we can train a generative model for the clean data (and with good enough architecture, inductive biases, having a small number of samples from the data would be sufficient). But if this breaks in practice, doesn't that mean that the theory is insufficient? While it makes sense that there is a gap between theory and practice, here the gap is "cheating" by incorporating clean samples, which undermines the whole point of the identifiability condition. If we need to use clean data for SFBD-OMNI, what's the difference between SFBD-OMNI and, e.g., Ambient Diffusion Omni [1] (which, to my knowledge, doesn't even require access to the operator)? Correct me if I am wrong, but I do not see a comparison with this method (even though it appears in your citations and related work section). Is this because it is very recent (NeurIPS 2025)?
> >
> > Unfortunately, my primary concerns are not yet fully addressed. I am looking forward to continuing the discussion with the authors.
> >
> > [1] G Darras et al. Ambient Diffusion Omni: Training Good Models with Bad Data. In NeurIPS 2025.

---

> ### Author Response · Authors · 2025-11-26
> **Thank you very much for the reply**
>
> Thank you very much for your detailed reply and we appreciate that you are willing to continuing the discussion.
>
> **Q5.** Regarding the requirement on a small number of clean samples.
>
> **A5.** First of all, we do not believe that assuming access to a very small number of clean samples undermines the validity of our algorithm or theory. This data regime is common in many real-world applications. In medical imaging, for example, high-quality, artifact-free scans require long acquisition times or high radiation doses, so clinicians typically collect only a few clean scans from volunteers or phantoms, while the vast majority of routine scans contain noise, motion blur, undersampling artifacts, or compression. Likewise, in satellite imaging, most observations are degraded by atmospheric effects, cloud cover, or low-light conditions, and only a small fraction of satellite passes yield clean images. Thus, the assumption of having a small clean dataset and a large corrupted dataset is both realistic and practically relevant, and the proposed method is well aligned with the needs of these applications.
>
> *Regarding why identifiability conditions matter*, we emphasize that although both scenarios assume access to a small number of clean samples, the way these samples are used differs fundamentally. When the identifiability condition holds, clean samples are required only to initialize the model (i.e., to set $p_0$ ) for faster convergence, and discontinuing their use in subsequent SFBD-OMNI iterations yields the best performance (see Fig. 2, CIFAR-10 with additive Gaussian noise). In contrast, when identifiability fails, clean samples must continue to be incorporated during iterative optimization by choosing p > 0. Otherwise, the model converges to an arbitrary element of the solution set $\mathcal{S}(q)$, leading to significantly degraded performance. This behavior is evident in Fig. 2 for CelebA with Grayscale and Gaussian Blur corruptions: the FID increases from 11.21 to 25.02 and from 10.28 to 76.30, respectively, when p is reduced from 0.2 to 0. The identifiability condition provides a clear and principled explanation for this phenomenon.
>
> *Regarding the gap between theory and practice*, we note that most of our theoretical results assume access to the corrupted density $q = \mathcal{T}\_r p\_{\rm data}$, which implicitly requires infinitely many corrupted samples. Under this idealized assumption, one can in principle recover $p_{\text{data}}$ without any clean samples whenever the identifiability condition holds. Conversely, if identifiability fails, then no amount of corrupted data, finite or infinite, can enable recovery in the absence of clean samples. In this sense, the apparent "gap" between theory and practice in the identifiable case—namely, the need for a small number of clean samples—is not due to a flaw in the theory, but rather to the fact that, in practice, the available noisy samples are insufficient to accurately estimate $q$, and we discussed this in Sec 3.2.
>
> Finally, according to our theory, when the corruption function is identifiable, a small number of clean samples are needed only to obtain a good initial distribution $p_0$. This also implies that if clean samples from the target distribution are unavailable, it is acceptable to use samples from a similar distribution instead. To demonstrate this, we pretrain the model on CIFAR-10 using clean samples from the truck class, and then apply iterative optimization to recover the distributions of automobile, ship, and horse, where all samples are corrupted by additive Gaussian noise with noise level $\sigma = 0.2$. The FID scores before and after iterative optimization are shown below:
>
> | Class               | After Pretrain | Final  |
> |---------------------|----------------|--------|
> | Automobile          | 8.36           | 6.19   |
> | Ship                | 13.96          | 8.78   |
> | Horse               | 25.87          | 13.55  |
> | Horse (no pretrain) | —              | 80.17  |
>
> As the table shows, for classes similar to truck -- such as automobile -- the model successfully recovers the target distribution, as indicated by the low final FID. For classes that are less similar, pretraining still provides substantial benefits. In particular, for horse, pretraining on the truck class reduces the final FID dramatically from 80.17 (without pretraining) to 13.55, illustrating the importance of a good initial distribution even when the clean samples are drawn from a different -- but related -- class. (Notably, the horse and truck classes still share several low-level characteristics such as edges and common background elements like grass or road surfaces.) We will include a summary of our discussion in the revision.

---

> ### Author Response · Authors · 2025-11-26
> **Thank you very much for the reply (cont'd)**
>
> **Q6** Regarding the work -- Ambient Diffusion Omni (Ambient-o)
>
> **A6.** We agree that this recent work is relevant and we will include a discussion and benchmark comparison in the revision. In particular, Ambient-o incorporates corrupted samples by injecting additional Gaussian noise. The key idea is that once sufficient Gaussian noise is added, the corrupted-noisy distribution and the clean-noisy distribution become harder to distinguish. This observation suggests that a corrupted sample, after being further noised, can effectively be treated as a noised clean sample, allowing it to be used in standard diffusion-model training. (A similar idea was also leveraged in the well-known SDEdit [1].) This effect does not depend on the specific form of the corruption process, allowing AD-OMNI to operate without requiring knowledge of the corruption mechanism.
>
> However, this strategy comes with an inherent trade-off. While heavy noising helps align corrupted samples with clean ones, it also risks erasing useful structure and details within the observations. In other words, sufficient noise is needed for Ambient-o to function as intended, but excessive noise may suppress informative signal that could otherwise benefit model learning.
>
> In contrast, SFBD-OMNI does not inject additional noise into the samples and therefore preserves the full information of the observations. Rather than relying on excessive noising to align distributions, our method leverages knowledge of the corruption process itself, avoiding information destruction while still enabling effective training.
>
> We include the following CIFAR-10 FID results in the revision. Following the setting of Ambient-o, we apply Gaussian-blurred data at varying blur strengths (σ) and assume access to 10% of clean samples.
>
> | Blur Strength (Gaussian σ) | Ambient-o | SFBD-OMNI |
> |----------------------------|-----------|-----------|
> | 0.6                        | 5.34      | 0.97      |
> | 1.0                        | 6.16      | 3.07      |
>
> As shown above, without injecting additional noise, SFBD-OMNI achieves substantially lower FID across blur levels, outperforming Ambient-o by a large margin.
>
> **Q7** Regarding the satellite data with Poisson noise.
>
> **A7.** Thank you for the detailed feedback and understanding. We agree that the current Poisson-noise results exhibit noticeable artifacts and are not yet satisfactory for practical applications. To ensure a balanced and realistic evaluation of our method, we will clearly acknowledge this limitation in the revised manuscript. We also appreciate your suggestion to mention failure modes rather than focusing solely on successful cases, and we will incorporate a transparent discussion of this limitation to more accurately reflect the current scope and robustness of our approach.
>
> Following your suggestion, we are currently running additional experiments with higher levels of Poisson noise that introduce more severe corruption. Although training is still ongoing, preliminary results indicate a clear trend: reconstruction quality declines as the noise intensity increases. We will report the completed results and include them in the revision, and include side-by-side visualizations of the ground-truth images and reconstructed outputs to better illustrate the behaviour of SFBD-OMNI under different corruption severities.
>
> [1] SDEdit: Guided Image Synthesis and Editing with Stochastic Differential Equations. Chenlin Meng, Yutong He, Yang Song, Jiaming Song, Jiajun Wu, Jun-Yan Zhu, Stefano Ermon, 2021

---

> > ### Author Response · Authors · 2025-11-27
> >
> > Dear reviewer,
> >
> > For your information, we have updated the revision to include the additional responses and empirical results. In particular:
> >
> > 1.	**Appendix K**: We added further empirical results on satellite data with Poisson noise at varying noise levels. We also expanded our discussion to highlight the limitations and drawbacks of the current implementation, with corresponding remarks referenced in the main text.
> >
> > 2.	**Appendix I**: We have incorporated additional results on Ambient Diffusion Omni (Ambient-o) and expanded the related work discussion in **Appendix B**.
> >
> > 3.	We have integrated the remaining discussion into Section 6, and included results for models pretrained on related datasets in **Appendix J**.
> >
> > We would be glad to discuss further if additional clarification would be helpful.

---

> > > ### Comment · Reviewer_Sh1d · 2025-11-27
> > > **Thank you!**
> > >
> > > I appreciate the authors’ efforts to address my concerns, and I recognize that they have effectively integrated our discussion into the paper. While I find this work interesting, I am still not convinced that the requirement for a small number of clean samples should not be considered a major limitation. Although some real-world applications may indeed allow access to a limited set of clean samples, new experiments on satellite data with Poisson noise (a relatively simple degradation and a relatively simple clean data distribution) indicate that even in such scenarios, the proposed algorithm can fail. Thus, the method does not appear to offer a clear advantage or particularly strong novelty over prior approaches such as Ambient Diffusion Omni. In my view, this is an incremental, though still very interesting, contribution. I thereby retain my original ratings.

---

### Official Review · Reviewer_6bRN · 2025-10-30

**Soundness:** 3
**Presentation:** 4
**Contribution:** 3
**Rating:** 6
**Confidence:** 3

**Summary:**

This paper considers the problem of restoring clean signals from corrupted measurements in a setting with little to no access to clean samples and a black-box access to the corruption model. Taking inspiration from the Stochastic Forward-Backward Deconvolution (SFBD) algorithm for Gaussian noise corruption, the authors introduce SFBD-OMNI as its extension to arbitrary corruption processes. At the core of the method lies a reformulation of the problem using the Donsker-Varadhan variational principle, which leads to an explicit objective for the (augmented) Kullback-Leibler Ambient Projection (KLAP) problem. In addition to the resulting formulation of SFBD-OMNI, the authors also propose its online (flow) variant that performs only partial updates to the data buffer used during training. Both methods are empirically compared with a proper set of baselines on both identifiable and non-identifiable processes. In addition, an ablation regarding the influence of the clean sample weighting is presented.

**Strengths:**

S1. The paper is written extremely well considering that the topic is not the easiest to describe formally. The flow of the text is great, with additional intuitive explanations following the introduction of more difficult concepts (like Figure 1).

S2. The authors propose a novel reformulation of the considered objective using the Donsker-Varadhan variational principle, showing a connection to the entropic optimal transport objective. The theoretical results are interesting and relevant to the considered problem. Importantly, the convergence analysis applies to both the standard SFBD-OMNI and the online version.

S3. The proposed methods outperform the considered baselines and the empirical results align with the theoretical considerations.

**Weaknesses:**

W1. My main concern relates to the accuracy of one of the claims. The authors suggest that, under suitable identifiability conditions, the ground-truth clean data distribution can be recovered without access to clean samples (as lines 065-069 seem to suggest). Proposition 4 guarantees convergence under the considered update rules, even with an arbitrary initialization and full replacement in the training buffer $\Large \varepsilon$. Does this imply that SFBD works when no clean samples are used at all, i.e., even when the initial pretraining phase is skipped? Is this scenario empirically verified in the paper, or do the results only cover the case where clean samples are used for pretraining but omitted from the updates in line 4 of Algorithm 1/2? If this is not verified, I would ask the authors to provide results for this case.

W2. I understand that some methods in Table 1 are only applicable to specific corruption processes. Are there any instances where a baseline is theoretically suitable for one of the tested problems, but its results are not included in the table? If so, the paper would greatly benefit from evaluating these baselines in such cases, as this would provide a more complete comparison and support future work.

**Questions:**

I am generally very sympathetic to this work and consider the approach, together with its theoretical derivations, very elegant. I was leaning toward a score of 8. However, I am very interested in the authors' answer to W1 above and consider it an important clarification before raising my score from 6 to 8. Please also refer to W2. Below are some additional minor questions.

Q1. lines 046-048: Is this phrased correctly? Shouldn't it be *With only a limited number of corrupted samples but an abundance of clean ones (...)*?

Q2. line 160: The authors probably meant $g=0$ for Eq. 6.

Q3. I-projection (line 216) should be expanded to Information-projection for clarity.

Q4. line 253: What if $r(\mathbf{y}|\mathbf{x})=0$ under the $\log$?

Q5. lines 289-298: The subscripts and superscripts for $k$ are probably mixed up.

---

> ### Author Response · Authors · 2025-11-23
>
> Thank you very much for your thoughtful and constructive feedbacks. We are encouraged by your recognition of our theoretical and emprical contribution. Our responses are as follows:
>
> **Q1.** *Regarding the identifiability conditions and no clean sample cases.*
>
> **A1.** Yes, your understanding is correct. When the corruption process satisfies the identifiability condition, SFBD-OMNI can, in principle, recover the true data distribution using only noisy samples. However, we emphasize that Proposition 4 assumes access to the true corrupted distribution, whereas in practice we only observe its empirical approximation constructed from finitely many noisy samples.
>
> In other words, Proposition 4 guarantees that, with infinitely many noisy samples, SFBD-OMNI can recover the clean data distribution under an identifiable corruption process. Unfortunately, for many common corruption processes, the sample complexity is extremely unfavorable, making it practically infeasible to collect enough noisy samples to train a high-quality model. In such cases, incorporating even a small number of clean samples can significantly mitigate this issue and lead to effective learning in practice. We have added a paragraph discussing this sample complexity issue at the end of Sec 3.2 to clarify this.
>
> Empirically, we have added Fig 3c in the revision to illustrate what happens when SFBD-OMNI is applied without any clean samples (see also Lines 511–519 for additional discussion). In short, the model can still approximately learn the data distribution, as evidenced by the steadily decreasing FID across iterations. However, the final FID remains substantially higher than in the setting where just 50 clean samples are provided, highlighting the practical benefit of incorporating a small amount of clean data.
>
>
> **Q2.** *Regarding the missing empirical results of the benchmarked models in Table 1.*
>
> **A2.** This is a good suggestion. In the revised version, we have added the missing SURE-Score results for the grayscale setting, as this corruption model is compatible with the assumptions required by SURE-Score. Other missing entries involve corruption models outside the valid operating regimes of these methods; we thus leave them blank accordingly.
>
> **Q3** *Other minor issues.*
>
> **A3.** We have corrected the typos you noted in the revised version. Regarding the phrase "a limited number of clean samples but an abundance of corrupted ones" we believe this is accurate. Our setting does not assume access to a large clean dataset; otherwise, one could simply train a generative model directly on the clean samples. Instead, our goal is to leverage a small clean set together with a much larger set of corrupted observations. For $r(\mathbf{x} | \mathbf{y})$ now at Line 260, we have added a footnote to clarify that "We assume $r( \cdot \mid \mathbf{x})$ has full support; this can be enforced by injecting an infinitesimal Gaussian noise to $\mathbf{y}$."

---

> > ### Comment · Reviewer_6bRN · 2025-11-26
> >
> > Thank you for addressing my concerns. After considering the paper in its current state, I have decided to maintain my score. I believe it accurately reflects the level of novelty and the performance demonstrated by the proposed method.
> >
> > One additional comment: in the considered setting, one could argue that _a limited number of clean samples_ + _access to the corruption model_ effectively implies _access to corrupted samples_. Therefore, I still find the phrasing 'a limited number of clean samples but an abundance of corrupted ones' to be misleading in this context.
> >
> > I remain open to any additional results that could highlight further novelty or performance gains to reconsider the score.

---

> ### Author Response · Authors · 2025-11-26
>
> Thank you very much for the reply. We are glad to learn that our responses address the concerns.
>
> We are not so sure if we understand your additional comment precisely. To avoid misunderstanding, we would like to clarify our intended setting. In our problem formulation, corrupted samples are assumed to come from real observations rather than being generated synthetically through the corruption model. The phrase "limited clean samples but abundant corrupted ones: is meant to reflect practical scenarios where only a small clean dataset is available, yet corrupted observations occur frequently and at scale (e.g., due to imperfect acquisition pipelines or sensor degradation).
>
> Although we agree that, in principle, having access to the corruption model enables one to generate corrupted samples from clean data, a limited number of clean samples + access to the corruption model would only yield a limited number of synthetically corrupted samples. In contrast, our assumption is that corrupted data are plentiful in the real world, independent of the small clean subset -- which is why we describe the regime as "clean sample-scarce but corrupted one-rich."
>
> We would be glad to discuss this further if additional clarification would be helpful. In addition, in **Appendix J** of the updated revision, we provide results and discussion showing that when the identifiability condition is satisfied, one may pretrain the model using related datasets when no samples from the target domain are available.

---

### Official Review · Reviewer_GiSu · 2025-10-31

**Soundness:** 2
**Presentation:** 2
**Contribution:** 2
**Rating:** 2
**Confidence:** 3

**Summary:**

The authors propose SFBD-OMNI, a bridge model–based framework that maps corrupted sample distributions to the underlying clean data distribution for image restoration and generation recovery. The method extends the prior SFBD framework to handle arbitrary measurement models beyond the Gaussian corruption setting, providing a more general and theoretically grounded approach to learning from imperfect or partially observed data

**Strengths:**

- Theoretical generalization capability for any unbound noise. The paper extends prior diffusion-based deconvolution frameworks SFBD to non-identifiable measurement processes, providing a principled formulation via one-sided entropic optimal transport. This generalization makes the approach applicable to a broader range of real-world degradations
- Data-efficient restoration without extensive ground-truth supervision. The proposed framework learns to recover clean image distributions using only a handful of clean samples, making image restoration more practical and label-efficient compared to traditional fully supervised diffusion approach

**Weaknesses:**

- Limited novelty via SFBD framework extension. While the paper provides a principled generalization of SFBD to non-identifiable measurement models via one-sided entropic OT, the overall formulation and optimization pipeline remain close to the original SFBD framework. The methodological increment feels more like an extension or refinement rather than a fundamentally new paradigm.

- Insufficient experimental evidence to substantiate the theoretical analysis. Despite a well-developed theoretical formulation, the experimental validation is narrow and mostly relies on simple synthetic corruption processes. The work lacks evaluation on real-world restoration datasets (e.g., **RainDrop (Qian et al., CVPR 2018)**, **AllWeather (Valanarasu et al., 2022)** ) that could better demonstrate generalization under uncontrolled degradations. Moreover, the notion of “limited clean data” (only 50 samples) is fixed and not systematically analyzed against traditional fully-supervised baselines, leaving unclear how much clean supervision is truly needed.

- Sensitivity to hyperparameter settings and practical complexity:
The algorithm involves several critical hyperparameters, such as the clean-sample weight λ and the online update ratio γ, which the authors fix empirically (λ / (1 + λ) = 0.2, γ = 0.002). The paper does not provide an ablation or sensitivity study, and training remains complex due to alternating optimization of all hyperparameters. This raises concerns about reproducibility and the need for manual tuning to achieve stable performance.

**Questions:**

1.	How does the method scale with different amounts of clean data?
The paper fixes the number of clean samples to 50 (~0.1%), but it would be informative to see how performance changes when this ratio varies, or how it compares to a small supervised diffusion baseline trained on the same subset?

2.	Do the authors plan to evaluate SFBD-OMNI on real-world restoration datasets (e.g., RainDrop, AllWeather) to demonstrate its applicability beyond synthetic corruptions, and to compare its performance against recent fully supervised diffusion-based restoration methods?

---

> ### Author Response · Authors · 2025-11-23
>
> Thank you for your feedback. Our responses are as follows:
>
> **Q1.** *Regarding novelty.*
>
> **A1.** The original SFBD algorithm hinges on the observation that the forward diffusion process can be interpreted as an additive Gaussian corruption of the data. This viewpoint naturally leads to a training framework tailored to learning from Gaussian-corrupted samples. In contrast, SFBD-OMNI is motivated by an alternative variational formulation of the KLAP problem that does not depend on diffusion dynamics. Although the original SFBD can be recovered as a special case of SFBD-OMNI, the two methods arise from very different perspectives. This distinction allows our formulation to naturally accommodate arbitrary corruption processes, which we view as a clear and non-trivial contribution both theoretically and empirically. In addition, we introduce an online variant of the method together with convergence results -- an aspect that, to the best of our knowledge, has no comparable counterpart in the existing ambient-related literature.
>
>
> **Q2.** *Regarding the emprical results on real world datasets.*
>
> **A2.** The datasets we selected are consistent with those used in the existing ambient-related literature (e.g., SFBD). In response to the requests for more realistic restoration experiments, we have included additional empirical results in Appendix I. These new experiments cover satellite imagery corrupted by Poisson noise and MRI data degraded by compressive-sensing operators. We further elaborate on why these corruption models are natural and widely used for the corresponding modalities.
>
> Across these higher-resolution and more practically relevant settings, the observed performance trends remain consistent with our results on CIFAR-10 and CelebA. These findings provide additional support for our theoretical claims and further demonstrate the robustness and practical effectiveness of SFBD-OMNI.
>
> **Q3.** *Regarding the sensitivity study on hyperparameters*
>
> **A3.** In the revision, we have added ablation studies on the number of clean samples and on the online update ratio $\gamma$ in Sec 6 (see Fig 3). Together with the existing analysis on the clean-sample weighting, we hope these results provide a more comprehensive understanding of the behavior and design choices of SFBD-OMNI.

---

> > ### Comment · Reviewer_GiSu · 2025-11-25
> >
> > Thanks for the detailed and constructive author feedback. After reviewing all the clarifications, the additional explanations, and the new ablation studies, I have recalibrated my understanding and updated my score accordingly.

---

> > > ### Author Response · Authors · 2025-11-26
> > >
> > > Thank you for your follow-up. We are pleased that the clarifications and new experiments helped address the concerns and contributed to a more complete understanding of the work.

---

### Official Review · Reviewer_b1RA · 2025-11-04

**Soundness:** 3
**Presentation:** 1
**Contribution:** 3
**Rating:** 6
**Confidence:** 5

**Summary:**

The paper proposes SFBD-OMNI, a bridge-model framework to learn the clean data distribution when you have many corrupted samples and few clean ones, assuming black-box access to the corruption process. They cast “recover clean from corrupted” as a KL Ambient Projection (KLAP) problem and show two key variational views: (i) the classic AmbientGAN min–max formulation, and (ii) a new formulation via the Donsker–Varadhan principle that turns KLAP into a one-sided entropic optimal transport objective with a tractable alternating minimization (posterior update + prior update).

This yields closed-form updates and naturally plugs into diffusion/bridge models to learn the posterior
 $u_\theta(x∣y)$. To address non-identifiable corruption (e.g., grayscale or blur), they augment the objective with a KL regularizer toward a small set of clean samples $h$ and analyze the limit as the regularization weight vanishes, recovering the I-projection within the feasible set. They present SFBD-OMNI and an online variant (partial refresh of reconstructed samples) with convergence guarantees. Experiments on CIFAR-10 and CelebA show improved FID over baselines (including in non-identifiable cases) when a small number of clean samples is mixed in training.

**Strengths:**

- Unifying view + derivation: Recasting KLAP with Donsker–Varadhan to obtain a one-sided entropic OT objective is elegant and principled; it clarifies when/why alternating updates should work and connects to entropic OT literature (contrast with AmbientGAN’s min–max).
- Identifiability analysis + “few clean” fix: Clear conditions for identifiability and a regularized KL that provably picks the best element in the feasible set, matching practice where a handful of clean samples are available.
- Convergence guarantees: Both batch and online SFBD-OMNI have convergence results (including a rate bound when $\lambda \to 0$). That’s stronger theory than many EM-like or Ambient-style methods.
- Empirical coverage: Results span identifiable (masking, Gaussian noise) and non-identifiable (grayscale, blur) regimes; OMNI competes with or beats specialized methods and EM-Diffusion.
- Practical modeling choice: Using bridge/flow-matching to learn posteriors is a strong fit; the paper also addresses degeneracy of deterministic paths with noise injection.

**Weaknesses:**

- Assumption on black-box sampling from r(y|x): Many real sensors provide only forward passes on x, not easy sampling of y|x across conditions; measuring cost or calibration drift may complicate the assumed operator access. The paper could discuss robustness to operator mismatch (misspecified r).
- Posterior learning scalability/details: Training $u_\theta(x|y)$ as a conditional bridge can be heavy; stability hinges on path choices and endpoint noise. More ablations on bridge choices, noise level, and compute vs. FID trade-offs would help (training time is non-trivial).
- Evaluation breadth: Only CIFAR-10 ($32^2$) and CelebA ($64^2$) with limited corruptions. Stronger evidence would include higher-res datasets, complex forward models (e.g., compressive sensing, Poisson noise), and domain tasks where distributional recovery helps downstream reconstructions.
- FIDs are fine but you really do need to show empirical samples for the learned model.
- Tuning $\lambda, \gamma$: While theory supports limits, practice shows a “sweet spot” for clean-weight; guidance for automatic selection (e.g., validation via corrupted likelihood) is missing.
- Identifiability test practicality: The paper discusses conditions and a criterion qualitatively; a procedural test practitioners can run on a new corruption (and with finite data) would increase usability.
- Minor nitpick, but Section 4.2 needs some cleaning up bc there's too much content in a small space and it's also a key contribution. Maybe reduce some content in Section 4.1 to give more space to 4.2? Also a small section with notation may be helpful, I had to search for the definition of $h$ which was last defined in Section 3.2 (maybe just say $p_data$?). Also, between Eqn (14) and (15), can you add a small phrase saying "taking \mathbb{E} over $y\sim q$ and subtracting E[ log q(y) ], we get..." It was a little confusing at first since there are many functions and it wasn't clear what's a function of y, x, etc.

**Questions:**

- Operator misspecification: How sensitive is SFBD-OMNI to errors in $r(y|x)$? Say you're doing pixel dropout and you have the wrong estimate of dropout probabilities?
- Automatic hyper-selection: How exactly do you select the $\lambda$ (clean-weight) and $\gamma$ (online update ratio) parameters?
- For my own understanding, what's the right way to think of $\Phi(p)$ in Proposition 3? Is it a penalty for how different the pushforwards in y are for $p^*$ and $p$?

---

> ### Author Response · Authors · 2025-11-23
>
> Thank you for your positive feedback and for acknowledging our theoretical contributions. Our responses are as follows:
>
> **Q1**. *Regarding the empirical study on higher-resolution dataset and more complex corruption processes.*
>
> **A1**. We have added additional empirical results in Appendix I, including experiments on satellite images with Poisson noise and MRI images with compressive-sensing corruption. We also provide further descriptions to justify why these corruption models are natural choices for the respective datasets.
>
> The empirical results on these higher-resolution and more realistic settings are consistent with our earlier findings on CIFAR-10 and CelebA, further corroborating both our theoretical claims and the practical effectiveness of SFBD-OMNI.
>
> **Q2**. *Regarding the samples generated by the models.*
>
> **A2**. In the revised version, we have included the generated samples in Appx H and Appx I.
>
>
> **Q3**. *Regarding the hyperparameter selections on $\lambda$ and $\gamma$.*
>
> **A3**. Regarding the choice of $\lambda$ in our experiments, the ablation results indicate that the model’s performance is not particularly sensitive to $p = \frac{\lambda}{1+\lambda}$ in the vicinity of $p = 0.2$, and that this setting yields consistently strong results. For this reason, we adopt $p = 0.2$ as the default value in our empirical study and use $p = 0$ only when the corruption process satisfies the identifiability condition (in which case we know the model will achieve better performance).
>
> In addition to the ablation study on the regularization parameter $\lambda$, we have also included an ablation on the reconstructed-sample update ratio $\gamma$ in Sec 6. The results show that although a large update ratio can accelerate early training, it causes the target distribution to shift too rapidly for the model to adapt, eventually destabilizing training and degrading sample quality. This issue can be mitigated by choosing a smaller $\gamma$ (at the cost of slower convergence). For this reason, we adopt $\gamma = 0.002$ as the default setting in our experiments.
>
> **Q4**. *Regarding the presentation of Sec 4.2.*
>
> **A4**. In the revision, we have made the corresponding modifications according to your suggestions.
>
> **Q5**. *Regarding the corruption function misspecification.*
>
> **A5**. We acknowledge that this is an important point and that it can indeed pose a significant challenge in many practical scenarios. At the same time, we would like to emphasize that in many real-world applications, the corruption process is known or engineered (e.g., the compressive sensing operator in MRI), or can be estimated with high precision. Since our work focuses on developing a principled theoretical framework for distribution recovery via diffusion-based models, under the assumption that the corruption function is available as a black-box generator, we believe it is most appropriate to center the empirical study on validating our theoretical claims and understanding the behavior of the proposed method. Nonetheless, a detailed investigation of specific relaxations of this assumption would certainly be valuable and of independent interest, and we view it as a natural direction for future work that is complementary to the scope of the present paper.
>
> **Q6**. *Regarding $\Phi(p)$ in Prop 3.*
>
> **A6**. While $\Phi(p)$ is not itself a divergence between $q$ and the pushforward $\mathcal{T}_r p$, it serves as an entropic-OT relaxation of that mismatch: it quantifies how well a candidate distribution $p$ can jointly couple with the observed $q$ under the corruption model. In particular, it can be viewed as the minimum entropic transport cost required to explain the observed distribution $q$ starting from the candidate clean distribution $p$ under the corruption model. The inner optimization searches for the most plausible joint coupling $\pi(x,y)$ where
> (i) endpoint: the $y$-marginal matches $q$, and
> (ii) transition: the coupling places mass on pairs $(x,y)$ that are consistent with the corruption model through the cost $c(x,y) = -\log r(y \mid x)$.
>
> In this sense, $\Phi(p)$ measures how much "effort" (in an entropic-OT sense) is needed to map $p$ to the observations $q$. The optimal coupling $\pi^*$ represents this least-cost explanation of the data. When $\lambda = 0$, any $p$ satisfying $\mathcal{T}_r p = q$ naturally meets both (i) and (ii), and therefore attains minimal cost in this formulation.

---

### Author Response · Authors · 2025-12-03
**Summary of Reviews and Post-Rebuttal Revisions**

Dear ACs and reviewers,

We sincerely appreciate your time and effort in handling our submission. Below, we provide a summary of the reviewers’ comments and describe how we have addressed each point in the revised manuscript.

## Part 1: Points raised initially

Our submission initially received scores of 6, 6, 6, and 2. All four reviewers agreed that the paper makes a non-trivial theoretical contribution and significantly generalizes prior work such as SFBD. Reviewers b1RA, 6bRN, and Sh1d (all with initial scores of 6) further acknowledged the strength of our contributions, noting that "recasting KLAP with Donsker–Varadhan to obtain a one-sided entropic OT objective is elegant and principled." They also emphasized that both the proposed method and its online variant are promising and well supported by strong theoretical guarantees.

The main points raised were:

(i) GiSu (initial score 2): proposed algorithms' novelty is unclear when compared to SFBD.

(ii) b1RA, GiSu, Sh1d: additional ablation studies on the hyperparmeters selections.

(iii) b1RA, GiSu, Sh1d: additional empirical studies in complex and practical domains (e.g., MRI and satellite data)

(iv) 6bRN, Sh1d: clarifications on why clean samples are needed when the identifiability conditions are satisfied.

(v) Sh1d: additional motivation and explanation for the selected architecture, as well as results for other bridge implementation choices beyond flow matching.

## Part 2: Our revisions & how we addressed reviewers’ concerns.

To address Question (i), we have further clarified our contribution in the response and revised the corresponding paragraph in the manuscript.

To address Question (ii), we have added all the requested studies in Sec 6 of the revision.

To address Question (iii), we have added high-resolution satellite and MRI images with Poisson and compressive sensing corruption in Sec K. (Sh1d noted that the results on satellite images with Poisson noise were initially unsatisfactory. We have improved the implementation, which now yields significantly better results. See Part 4 for details.)

To address Question (iv), we added a discussion in Sec. 3.2 on sample complexity and clarified the assumptions in our theoretical analysis to explain why, in practice, a small number of clean samples is still needed to overcome the pessimistic sample complexity. We also included experiments in Sec. J to show that when clean samples are unavailable, samples from a similar distribution remain effective.

For Question (v), we added clarification in Sec. 6 and Appx. C on why we adopt a flow-matching backbone. While exploring other bridge designs would be valuable, our empirical focus is on validating the core theoretical and algorithmic framework, and we plan to investigate broader options in future work.

## Part 3: Status before the rollback.

GiSu (initial score 2) acknowledged that the novelty concern was addressed and expressed satisfaction with the added ablation study and new empirical results on the satellite and MRI datasets, **raising the score** accordingly to reflect a positive view of the submission. The reply explicitly states this change.

------

Reviewers 6bRN and Sh1d decided to maintain their scores. In paricular, 6bRN raised an additional comment on

(vi) why a small number of clean samples + access corruption model not the same as accessing many corrupted samples

We clarified in the reply that having “a small number of clean samples + access to the corruption model” would only yield a small number of corrupted samples, which differs from our setting that requires many corrupted samples. We had no chance to continue the discussion before the rollback.

6bRN and Sh1d also raised

(vii) the insufficient performance of the models in the newly added experiments on satellite images (which we have tried to address in Part 4).

Sh1d in addition raised

(viii) We need to differentiate our work from the very recent Ambient-o method published at the NeurIPS 2025 workshop.

To address this, we added a new discussion in the related work section and additional empirical results in Appx I. In particular, Ambient-o must further corrupt the already corrupted samples before using them for training, whereas our method does not, allowing it to fully leverage the information contained in the corrupted observations. Our empirical results in Appx I demonstrate this clearly: under the same setting, our method obtains an FID of 0.97, whereas Ambient-o achieves 5.34. Sh1d was not fully convinced of this point prior to the rollback, partly due to issue (vii), which we were in the process of resolving when the rollback occurred.

------

Although reviewer b1RA (initial score 6) did not have the chance to reply, he/she raised similar points about hyperparameter ablations and practical empirical results. We followed these suggestions in the rebuttal and believe the revisions address most concerns.

---

> ### Author Response · Authors · 2025-12-03
> **Summary of Reviews and Post-Rebuttal Revisions (cont'd)**
>
> ## Part 4: Additional results we added after the rollback
>
> As reviewers 6bRN and Sh1d noted the insufficient performance on satellite images corrupted by Poisson noise, raising questions about the framework’s potential for more complex tasks, we conducted further investigation and found that the degradation was largely caused by the use of Stable Diffusion’s autoencoder (which was adopted for saving computational resources only). Specifically, the autoencoder had never been pretrained on satellite images, leading to significant distortion. To address this, we performed training directly in pixel space, and the model’s performance is now much more promising. In particular, regarding the FID score,
>
> original revision (latent)
>
> | Stage          | α = 10  | α = 50  | α = 100 |
> |----------------|---------|---------|---------|
> | After Pretrain | 111.37  | 83.15   | 48.63   |
> | Final Result   | 81.49   | 47.05   | 37.53   |
>
> current revision (pixel):
> | Stage          | α = 10 | α = 50 | α = 100 |
> |----------------|--------|--------|---------|
> | After Pretrain | 9.32   | 5.71   | 4.43    |
> | Final Result   | 7.11   | 4.13   | 3.40    |
>
> We have updated the revision to accommodate this change and hope the results are now promising and can address the reviewer's concerns about our method's potential.
>
>
> ## Part 5: Concluding remarks
>
> This work provides a unified perspective on training generative models from noisy samples. While GAN training in this setting can be interpreted as solving an alternative variational problem, we show that diffusion-based models can also be trained effectively by formulating a new, suitable variational problem. We further show that this perspective is closely connected to optimal transport, and that the resulting method naturally addresses non-identifiability issues that cannot be resolved within GAN-based formulations. We further show that SFBD emerges as a special case of our method--SFBD is restricted to Gaussian corruptions, whereas our framework handles arbitrary corruption models. Although our original submission already included strong empirical results on the benchmarks used in related work such as SFBD, reviewers raised concerns about performance on more complex real-world tasks and the need for additional ablations. Reviewers have since acknowledged the adequacy of the ablation studies, and we believe the new results in Part 4 substantially address the remaining concerns about broader applicability. We hope this summary helps with the AC's recommendation.

---

### Meta-Review · Area_Chair_iTnT · 2026-01-21

**Summary:**

This paper proposes a bridge model that learns to reverse a (known) corruption process. The reviewers were generally positive about the theoretical contribution, especially the generalization of SFBD to arbitrary corruption models.

There are a few common concerns by the reviews. Three reviewers mentioned the proposed method's reliance on a small set of clean samples, which limits the method's applications to many real-world scenarios. The reviewers also raised concerns about the performance on more complex domains and using non-trivial corruption methods.

There were also discussions regarding the differences with SFBD, which the proposed method heavily based on, and a recent method Ambient-o. Reviewer Sh1d maintained that without a clear advantage in the "no clean sample" regime.

**Reviewer Concerns:**

Regarding the novelty with related methods, the authors clarified that SFBD-OMNI is derived from a different variational perspective allowing for arbitrary corruptions.

The authors provided additional experimental results on other datasets and different corruption methods to demonstrate the generalizability of the proposed method.

The reliance on clean samples remains, though the authors made the point that it is fundamentally impossible to reverse the corruption without prior knowledge on either the corruption type or some clean data.

**Reviewer Scores:**

Reviewer GiSu might increase their score as the authors addressed their main concerns in the rebuttal.

The other reviewers will likely maintain their score.

---

### Decision · Program_Chairs · 2026-01-26

Accept (Poster)